# Finding Safe Zones of Markov Decision Processes Policies

**Lee Cohen**
TTI-Chicago

**Yishay Mansour**
Tel-Aviv University
Google Research

**Michal Moshkovitz**
Bosch Center for AI

## Abstract

Given a policy of a Markov Decision Process, we define a SAFEZONE as a subset of states, such that most of the policy's trajectories are confined to this subset. The quality of a SAFEZONE is parameterized by the number of states and the escape probability, i.e., the probability that a random trajectory will leave the subset. SAFEZONES are especially interesting when they have a small number of states and low escape probability. We study the complexity of finding optimal SAFEZONES, and show that in general, the problem is computationally hard. Our main result is a bi-criteria approximation learning algorithm with a factor of almost 2 approximation for both the escape probability and SAFEZONE size, using a polynomial size sample complexity.

## 1 Introduction

Most research in reinforcement learning (RL) deals with learning an optimal policy for some Markov Decision Process (MDP). One notable exception to that is *Safe RL* which addresses the concept of safety. Traditional Safe RL focuses on finding the best policy that meets safety requirements, typically by either adjusting the objective to include the safety requirements and then optimizing for it, or incorporating additional safety constraints to the exploration. In both of these cases, the safety requirements should be pre-specified. *Anomaly Detection* is the problem of identifying patterns in data that are unexpected, i.e., anomalies (see, e.g., Chandola et al. (2009) for survey). This paper introduces the SAFEZONE problem, which addresses the safety of a specific Markov decision process (MDP) policy by detecting anomalous events rather than finding a policy that satisfies some pre-defined safety constraints.

Consider a finite horizon MDP and a policy (a mapping from states to actions). The policy induces a Markov Chain (MC) on the MDP. Given a subset of states, a trajectory *escapes it* if at least one of its states is not in the subset. The *escape probability* of a subset is the probability that a randomly sampled trajectory will escape it. A SAFEZONE is a subset of states whose quality is measured by (1) its escape probability and (2) its size. If a SAFEZONE has low escape probability, we consider it *safe* (hence escaping is the anomaly). We emphasize that safety is policy-dependent and that different policies could have different SAFEZONES.

Trivial solutions for SAFEZONE include the entire states set (minimal escape probability of 0, maximal size), and the empty set (minimal size, maximal escape probability of 1). The goal is to find a SAFEZONE with a good balance: a relatively small size but still safe enough (small escape probability). More precisely, given an upper bound over the escape probability, $\rho > 0$, the goal of the learner is to find the smallest SAFEZONE with escape probability at most $\rho$ using trajectory sampling. We address an unknown environment, by which we mean no prior knowledge of the transition function or the policy used. The learner is only given access to random trajectories generated by the induced Markov chain. For many applications, if a small SAFEZONE exists, it is useful to find it.

37th Conference on Neural Information Processing Systems (NeurIPS 2023).

One such example is designing a policy for a smaller state space that performs well in most cases but is undefined for some states, or formally, imitation learning with compact policy representation Abel et al. (2018); Dong et al. (2019). Suppose a company would like to automatically generate a 'lite' edition of a software or an app (e.g., Microsoft Office Lite, Facebook Lite) that contains only part of the system's states, finding a SAFEZONE makes a lot of sense— capturing popular users' trajectories. If instead of finding SAFEZONE , one were to simply take the $90\%$ most popular states of users using Office, they might not include the state that allows for the precious option of saving, which emphasizes the importance of the problem.

Another motivation for the problem is autonomous vehicles and specifically infrastructure design for them. Even though a lot of the progress in the field of autonomous driving is credited to sensors installed on the vehicles, relying solely on the vehicles' sensors has its limitations (e.g., Yang et al. (2020)). In extreme weather, a vehicle might unintentionally deviate from the current lane and the vehicle sensors might not trigger a response in time. Vehicular-to-Infrastructure (V2I) is a type of communication network between vehicles and road infrastructures that are designed to fill the need for an extra layer of safety.[1] An important part of the V2I communication is based on Road Side Units (RSUs), sensors that are installed alongside roads. Together with the sensors that are installed on the vehicles, they span the V2I communication. As the resources for RSUs distribution are limited and their enhanced safety is key for V2I and the autonomous vehicle adaptation, distributing RSUs in states of a (good) SAFEZONE could enhance the safety of popular commutes efficiently. Namely, given data regarding commutes (trajectories) in an area, installing RSUs in its' SAFEZONE states will ensure increased safety measures of a major part of the commutes, from starting point to destination, and potentially increase the trust in the system. In addition, if regulation were to prevent people from making autonomous commutes outside of the SAFEZONE , having most of the autonomous commutes confined to the SAFEZONE implies that most commutes can still be driverless.

Another useful application is automatic robotic arms that assemble products. If something unusual happened during the assembly of a product, it might result in a malfunctioning product, and in that case, the operator should be notified (anomaly detection). On the other hand, it is not really autonomic if the operator is notified too frequently. If we find a 'safe enough' SAFEZONE, we can make sure that we notify the operator only in the rare event the production process (trajectory) escapes it, which means that something went wrong with the product assembly. Furthermore, if the SAFEZONE is small, the manufacturer can potentially test the SAFEZONE states and verify their compliance, ensuring that the majority of products are well constructed for a significantly lower testing budget.

Finally, the SAFEZONE problem can be viewed through the lens of explainable RL, where the goal is to explain a specific policy. SAFEZONE is a new post-hoc explanation of the summarization type Alharin et al. (2020). For example, for the autonomous vehicle infrastructure design, governments could explain to their citizens the design that was chosen.

Our results include approximation algorithms for the SAFEZONE problem, which we show is NP-hard. We are interested in a good trade-off between the escape probability of the SAFEZONE and its size. Our algorithms are evaluated based on two criteria: their approximation factors (w.r.t. the escape probability bound and the optimal set size for this bound), and their trajectory sample complexity bounds (e.g., Even-Dar et al. (2006)).

**Contribution**: In Section 2 we formalize the SAFEZONE problem. In Section 3, we explore naive approaches, namely greedy algorithms that select SAFEZONES based on state distributions and trajectory sampling. In addition, we show particular cases in which their solutions are far from optimal, either in terms of high escape probability or significantly larger set size. In Section 4 we design FINDING SAFEZONE, an efficient approximation algorithm with provable guarantees. The algorithm returns a SAFEZONE which is slightly more than twice the size and twice the escape probability compared to the optimal. While the main focus of this work is the introduction of the problem and the aforementioned theoretical guarantees, we do demonstrate the problem empirically, to provide additional intuition to the readers. In Section 5, we compare the performance of the naive approaches to FINDING SAFEZONE and show that different policies might lead to completely different SAFEZONES. In Appendix A, we show that the problem is hard, even for known environment

---

[1]The 'V2I Deployment Coalition' is an initiative by the U.S. Department of Transportation with the vision of "An integrated national infrastructure that provides the country a connected, safe and secure transportation system taking full advantage of the progress being made in the Connected and Autonomous Vehicle arenas." https://transportationops.org/V2I/V2I-overview

setting, namely even when the induced Markov chain is given, finding a SAFEZONE is NP-hard, even for horizon $H = 2$.

For brevity, some algorithms and (full) proofs are relegated to the appendix.

## 1.1 Related Work

MDPs have been studied extensively in the context of decision making in particular by the Reinforcement Learning (RL) community (see Puterman (1994) for a broad background on MDPs, and Sutton & Barto (2018) for background on reinforcement learning).

**Safe RL** A related line of research is safe RL, where the goal of the learner is to find the best policy that satisfies safety guarantees. The two main methodologies to handle such problems are: (1) altering the objective to include the safety requirement and optimizing over it, and (2) adding additional safety constraints to the exploration part. See Pfrommer et al. (2021); Emam et al. (2021); Xu et al. (2021); Hendrycks et al. (2021); HasanzadeZonuzy et al. (2021); Bennett et al. (2023); Prajapat et al. (2022); Liu et al. (2022) for recent works and García & Fernández (2015); Amodei et al. (2016) for surveys. Recent work by Sootla et al. (2022) augments the environment to accommodate some pre-specified safety constraints and thus satisfies them almost surely. In our work, the goal is not to find the optimal policy, but rather, given a policy, finding its SAFEZONE. The SAFEZONE is not characterized by specific requirements, and might not be unique. Moreover, beyond the MDP, the solution very much depends on the policy.

**Imitation Learning.** In imitation learning, the learner observes a policy behavior and wants to imitate it (see Hussein et al. (2017) for a survey). Similar to imitation learning, we are given access to samples of a given policy. In contrast, rather than imitating the policy we find the policy's SAFEZONE, which is an important property of the policy.

**Approximate MDP equivalence.** Another related research line is that of finding an (almost) equivalent minimal model for a given MDP, where the goal is that the optimal policy on the (almost) equivalent model induces an (approximately) optimal policy in the original MDP, e.g., Givan et al. (2003); Even-Dar & Mansour (2003). This line of works and ours differ in that we do not try to modify the MDP (e.g., cluster similar states), but rather to find a SAFEZONE, a property that is defined for the existing MDP and a specific policy.

**Explainability.** In explainability, the goal is to provide a post-hoc explanation to a specific (given) model Molnar (2019), e.g., using decision trees Blanc et al. (2021); Moshkovitz et al. (2021), influential examples Koh & Liang (2017), or local approximation explanations Li et al. (2020). We focus on explainability for reinforcement learning, and specifically, we suggest a new summarization explanation through our SAFEZONE (Amir & Amir, 2018).

## 2 The Safe Zone Problem

We model the problem using a Markov model with a finite horizon $H > 1$. Formally, there is a Markov chain (MC) $\langle \mathcal{S}, P, s_0 \rangle$ where $\mathcal{S}$ is the set of states, $s_0 \in \mathcal{S}$ is the initial state, and $P : \mathcal{S} \times \mathcal{S} \to [0, 1]$ is the transition function that maps a pair of states into probability by $P(s, s') = \Pr[s_{t+1} = s' | s_t = s]$. We assume the transition function $P$ is induced by a policy $\pi : \mathcal{S} \to \text{Simplex}^{\mathcal{A}}$ on an MDP $\langle \mathcal{S}, s_0, P', \mathcal{A} \rangle$ with transition function $P' : \mathcal{S} \times \mathcal{A} \times \mathcal{S} \to [0, 1]$ such that $P(s, s') = \sum_{a \in \mathcal{A}} P'(s, a, s') \cdot \pi(a|s)$ for all $s, s' \in \mathcal{S}$ (though any MC can be generated this way, thus our theoretical guarantees apply for general MCs).

A *trajectory* $\tau = (s_0, \ldots, s_H)$ starts in the initial state $s_0$ and followed by a sequence of $H$ states generated by $P$, i.e., $\Pr[s_{i+1} = s' | s_i = s] = P(s, s')$ for all $i \in [H]$, where $[H] := \{1, \ldots, H\}$. For example, in the context of autonomous vehicles, the trajectory is a commute. We abuse the notation and regard a trajectory $\tau$ both as a sequence and a set.

Given a subset of states $F \subseteq \mathcal{S}$, a trajectory $\tau$ *escapes* $F$ if it contains at least one state $s \in \tau$ such that $s \notin F$, i.e., $\tau \nsubseteq F$. For example, if a commute passes through at least one area (state) that does not have an RSU sensor, it escapes the SAFEZONE . We refer to the probability that a random trajectory escapes $F$ as *escape probability* and denote it by $\Delta(F) = \Pr_\tau[\tau \nsubseteq F]$. We call $F$ a $\rho-safe$ (w.r.t. the model $\langle \mathcal{S}, s_0, P \rangle$) if its escape probability, $\Delta(F)$, is at most $\rho$. Formally,

**Definition 2.1.** *A set $F \subseteq \mathcal{S}$ is $\rho-safe$ if $\Delta(F) := \Pr_\tau[\tau \nsubseteq F] \leq \rho$, where $\tau$ is a random trajectory.*

A set $F \subseteq \mathcal{S}$ is called $(\rho, k)-$SAFEZONE if $F$ is $\rho-$safe and $|F| \leq k$. Given a safety parameter $\rho \in (0, 1)$, we denote the smallest size $\rho-$safe set by $k^*(\rho)$:
$$k^*(\rho) = \min_{F \subseteq \mathcal{S} \text{ is } \rho-\text{safe}} |F|.$$
Whenever the discussed parameter $\rho$ is clear from the context we use $k^*$ instead of $k^*(\rho)$. We remark that there might be multiple different $(\rho, k)-$SAFEZONE sets. The learner knows the set of states, $\mathcal{S}$, the initial state, $s_0$, and the horizon $H$. However, the transition function $P$ and the minimal size of the $\rho-$safe set, $k^*$, are unknown to the learner. Instead, the learner receives information about the model from sampling trajectories from the distribution induced by $P$.

Given $\rho > 0$, the ultimate goal of the learner would have been to find a $(\rho, k^*(\rho))-$SAFEZONE. However, as we show in Appendix A, finding a $(\rho, k^*(\rho))-$SAFEZONE is NP-hard, even when the transition function $P$ is known. This is why we loosen the objective to find a bi-criteria approximation $(\rho', k')-$SAFEZONE . (Bi-criteria approximations are widely studied in approximation and online algorithms Vazirani (2001); Williamson & Shmoys (2011).) In our setting, given $\rho$ the objective is to find a set $F$ which is $(\rho', k')-$SAFEZONE with minimal size $k' \geq k^*$ and minimal escape probability $\rho' \geq \rho$. In addition, we are interested in minimizing the sample complexity.

Notice that the learner can efficiently verify, with high probability, whether a set $F$ is approximately $\rho-$safe or not, as we formalize in the next proposition. The following proposition follows directly from Lemma C.2.

**Proposition 2.2.** *There exists an efficient algorithm such that for every set $F \subseteq \mathcal{S}$ and parameters $\epsilon, \lambda > 0$, the algorithm samples $O(\frac{1}{\epsilon^2} \ln \frac{1}{\lambda})$ random trajectories and returns $\widehat{\Delta}(F)$, such that with probability at most $\lambda$ we have $|\Delta(F) - \widehat{\Delta}(F)| \geq \epsilon$.*

### 2.1 A Note on Trajectory Escaping

The SAFEZONE problem deals with escaping trajectories. In particular, given a SAFEZONE, a trajectory escapes it, no matter if only one of its states is outside the SAFEZONE or all of them. A related, yet very different problem, is that of minimizing a subset size, such that the expected number of states outside the set is minimized. This related problem, while significantly easier (as it is solved by returning the most visited states), does not apply to the applications we described earlier. For example, consider the infrastructure design for autonomous vehicles. We want passengers to have a safe experience end-to-end. Hence the entire route must have that extra security layer provided by the RSUs. In Section 3, we show that the solution for the SAFEZONE does not necessarily overlap with the most visited states. Furthermore, simply returning states that appeared in trajectory samples could result in a set size far from optimal.

### 2.2 A Note on Multiple Policies

Our framework can accommodate an arbitrary number of policies, representing multiple agents. This is made feasible through the implementation of a single mixed policy. This mixed policy is designed to stochastically select a policy from this ensemble of policies, uniformly at random or according to any specified distribution over the participating agents and their respective policies.

### 2.3 Summary of Contributions

We summarize the results of all the algorithms that appear in the paper in Table 1. The bounds of GREEDY BY THRESHOLD and GREEDY AT EACH STEP require the Markov Chain model as input, and a pre-processing step that takes $O(|S|^2 H)$ time. Additionally, the bounds for the first three algorithms (the naive approaches) require additional knowledge of $k^*(\rho)$. The sample complexities of SIMULATION is bounded by $poly(k^*, \frac{1}{\rho})$, and of FINDING SAFEZONE Algorithm is bounded by $poly(k^*, H, \frac{1}{\epsilon}, \frac{1}{\delta})$ for some parameters $\epsilon, \delta \in (0, 1)$.

Beyond the upper bounds, we provide each of the first three algorithms (the naive approaches) instances that show that they are tight up to a constant.

The following theorem is an informal statement of our main theorem, Theorem 4.2.

**Theorem 2.3.** *For every $\rho, \epsilon, \delta > 0$, with probability $\geq 0.99$ there exists an algorithm that returns a set which is $(2\rho + 2\epsilon, (2 + \delta)k^*) -$ SAFEZONE.*

In addition to the sample complexity, the running time of the algorithm is also bounded by $poly(k^*, H, \frac{1}{\delta}, \frac{1}{\epsilon})$.

We empirically evaluate the suggested algorithms on a grid-world instance (where the goal is to reach an absorbing state), showing that FINDING SAFEZONE outperforms the naive approaches. Moreover, we show that different policies have qualitatively different SAFEZONES. Finally, an informal statement of Theorem A.2 which appears in Appendix A due to space limitations.

**Theorem 2.4.** SAFEZONE *is NP-hard.*

Table 1: Upper bounds for safety and set size. * Only for layered MDPs.

| Algorithm | Safety | Set Size |
|---|---|---|
| Greedy by Threshold | $2\rho$ | $k^*H/\rho$ |
| Simulation | $2\rho$ | $O(k^*H \ln k^*)$ |
| Greedy at Each Step* | $\rho H$ | $k^*$ |
| Finding SAFEZONE | $2\rho + 2\epsilon$ | $(2+\delta)k^*$ |

## 3 Gentle Start

This section explains and analyzes various naive algorithms to the SAFEZONE problem. We show that even if the transition function is known in advance, these naive algorithms result in outputs that are far from optimal. To describe the algorithms, we define for each state $s$ the probability to appear in a random trajectory and denote it by $p(s) = \Pr_\tau[s \in \tau] \in [0, 1]$. Note that $\sum_{s \in \mathcal{S}} p(s)$ is a number between 1 and $H$ (e.g., $p(s_0) = 1$), and can be estimated efficiently using dynamic programming if the environment and policy are known and sampling otherwise. To be precise, some of the algorithms assume the probabilities $\{p(s)\}_{s \in \mathcal{S}}$ are received as input.

**Greedy by Threshold Algorithm.** The algorithm gets, in addition to $\rho$, the distribution $p$ and a parameter $\beta > 0$ as input. It returns a set $F$ that contains all states $s$ with probability at least $\beta$, i.e., $p(s) \geq \beta$. We formalize this idea as Algorithm 3 in Appendix B. For $\beta = \frac{\rho}{k^*}$, the output of the algorithm is $\left(2\rho, \frac{k^*H}{\rho}\right) -$ SAFEZONE. More generally, we prove the following lemma.

**Lemma 3.1.** *For any* $\rho, \beta \in (0, 1)$, *the* GREEDY BY THRESHOLD ALGORITHM *returns a set that is* $(\rho + k^*\beta, \frac{H}{\beta}) -$ SAFEZONE. *In particular, for* $\beta = \frac{\rho}{k^*}$, *this set is* $\left(2\rho, \frac{k^*H}{\rho}\right) -$ SAFEZONE.

While it is clear why there are instances for which the safety is tight, Lemma B.1 in Appendix B shows that the set size is tight as well.

**Simulation Algorithm.** The algorithm samples $O(\frac{\ln k^*}{\beta})$ random trajectories and returns a set $F$ with all the states in these trajectories. It is formalized in Appendix B as Algorithm 4.

**Lemma 3.2.** *Fix* $\rho, \beta \in (0, 1)$. *With probability at least* $0.99$, SIMULATION *Algorithm returns a set that is* $\left(\rho + k^*\beta, O(k^* + \frac{\rho H \ln k^*}{\beta})\right) -$ SAFEZONE. *In particular, for* $\beta = \frac{\rho}{k^*}$, *this set is* $(2\rho, O(k^*H \ln k^*)) -$ SAFEZONE.

While the algorithm achieves a low escape probability, only $2\rho$, in Lemma B.2 in the appendix we prove that the size of $F$ is tight up to a constant, i.e., we show an MDP instance where $|F| = \Omega(k^*H \ln k^*)$. The algorithms presented so far were approximately safe (i.e., low escape probability), but the returned set size was large. Without any further assumptions, the following algorithm provides a $(\rho H, Hk^*) -$ SAFEZONE, thus not improving the previous algorithms. However, when considering MDPs with a special structure it provides an optimal sized SAFEZONE , at the price of large escape probability.

**Greedy at Each Step Algorithm.** For the analysis of the next algorithm, we assume the MDP is *layered*, i.e., there are no states that appear in more than a single time step and denote $\mathcal{S} = \bigcup_{i=1}^{H} \mathcal{S}_i$. I.e., the transitions $P(s, s')$ are nonzero only for $s' \in \mathcal{S}_{i+1}$ and $s \in \mathcal{S}_i$. The GREEDY AT EACH STEP ALGORITHM, sometimes simply called greedy, takes at each time step $i$ the minimal number of states such that the sum of their probabilities is at least $1 - \rho$. It is formalized in Appendix B as Algorithm 5.

**Lemma 3.3.** *For any $\rho \in (0, 1)$, if the MDP is layered,* GREEDY AT EACH STEP ALGORITHM *returns a set that is* $(\rho H, k^*) -$ SAFEZONE.

In Lemma B.3 in the appendix we provide a lower bound on the escape probability, matching up to a constant.

**Weaknesses of the naive algorithms.** We showed algorithms that identify SAFEZONE with escape probability much greater than $\rho$ or size much greater than $k^*$, and instances with tight lower bounds for each of them. This holds even when providing extra information about the model or the optimal size of the $\rho-$safe set, i.e., $k^*$.

## 4 Algorithm for Detecting Safe Zones

In this section, we suggest a new algorithm that builds upon and improves the added trajectory selection of the SIMULATION Algorithm. One reason for why SIMULATION returns a large set is that it treats every sampled trajectory identically, regardless of how many states are being added, which could be as large as $H$. More precisely, fix any $(\rho, k^*) -$ SAFEZONE set, $F^*$, and consider a trajectory $\tau$ that escapes it, i.e., $\tau \not\subseteq F^*$. If $\tau$ was sampled, its states are added to the constructed set $F$, which might increase the size of $F$ by up to $H$ states that are not in $F^*$, without significantly improving the safety. In contrast, when selecting which trajectory to add to $F$, we would consider the number of states it adds to the current set. For the sake of readability, we refer to any state which is not in the current set $F$ as *new*, and denote by $new_F(\tau)$ the number of new states in $\tau$ w.r.t. $F$, i.e.,

$$new_F(\tau) := |\tau \setminus F|.$$

Note that for every $F \subseteq \mathcal{S}$, $\Pr_\tau[new_F(\tau) \neq 0] = \Delta(F)$.

The new algorithm does not sample each trajectory uniformly at random, but samples from a new distribution, which will be denoted by $Q_F$.

While favoring trajectories with higher probabilities, which we already get by the sampling process, another key idea would guide this new distribution: To prefer trajectories that *gradually* increase the size of $F$. To implement this idea, we will ensure that the probability of adding a trajectory $\tau$ to $F$ should be *inversely proportional* to $new_F(\tau)$.

Formally, the support of $Q_F$ is the trajectories with new states, i.e., $X = \{\tau | new_F(\tau) \neq 0\}$. For every $\tau \in X$

$$Q_F(\tau) \propto \frac{\Pr[\tau]}{new_F(\tau)},$$

where $\Pr[\tau]$ is the probability of trajectory $\tau$ under the Markov Chain with dynamics $P$. Note that the new distribution depends on the current set $F$, and changes as we modify it. Intuitively, adding trajectories to $F$ according to $Q_F$ instead of adding trajectories sampled directly from the dynamics (as we do in SIMULATION) would increase the expected ratio between the added safety and the number of new states we add to $F$, thus improving the set size guarantee of the output set. We elaborate on this in Section 4.2.

Our main algorithm is FINDING SAFEZONE, Algorithm 1. The algorithm receives, in addition to the safety parameter $\rho$, parameters $\epsilon, \lambda \in (0, 1)$, and maintains a set $F$ that is initiated to $\{s_0\}$. On a high level, to implement the idea of adding trajectories to $F$ according to $Q_F$, we use *rejection sampling*. Namely, in each iteration of the while–loop we first sample a trajectory $\tau$, and if $new_F(\tau) \neq 0$, we *accept* it with probability $1/new_F(\tau)$. If the trajectory is accepted, it is added to $F$. More precisely, if $new_F(\tau) \neq 0$, we sample a Bernoulli random variable, $accept \sim Br(1/new_F(\tau))$. If $accept = 1$, we add $\tau$ to $F$. This process of adding trajectories to $F$ generates the desired distribution, $Q_F$. Whenever a trajectory is added to $F$, we estimate the escape probability $\Delta(F)$ (w.r.t. the updated set, $F$).

The algorithm stops adding states to $F$ and returns it as output when it becomes "safe enough". To be precise, let $\widehat{\Delta}(F)$ denote the result of the escape probability estimation (by sampling trajectories as suggested in Proposition 2.2). If $\widehat{\Delta}(F) \leq 2\rho + \epsilon$ , it means that $F$ is $(2\rho + 2\epsilon)-$safe with probability $\geq 1 - \lambda_j > 1 - \lambda$, in which case the algorithm terminates and returns $F$ as output.

To implement the estimation $\widehat{\Delta}(F)$, the algorithm calls *EstimateSafety* Subroutine. The subroutine samples $N_j = \Theta(\frac{1}{\epsilon^2} \ln \frac{2}{\lambda_j})$ trajectories, and returns the fraction of trajectories that escaped $F$.

For cases in which the transition function $P$ is known to the learner, we provide an alternative implementation for *EstimateSafety* which computes the exact probability $\Delta(F)$ (see Lemma E.1 in Appendix E).

---

**Algorithm 1** FINDING SAFEZONE

  Input: $\rho \in (0,1)$
  Parameters: $\epsilon, \lambda \in (0,1)$
  $F \leftarrow \{s_0\}, j \leftarrow 1, \widehat{\Delta}(F) \leftarrow 1$
  **while** $\widehat{\Delta}(F) > 2\rho + \epsilon$ **do**
    $\tau \leftarrow$ sample a random trajectory
    Compute $new_F(\tau)$
    **if** $new_F(\tau) \neq 0$ **then**
      sample $accept \sim Br(1/new_F(\tau))$
      **if** $accept = 1$ **then**
        $F \leftarrow F \cup \tau$
        $\lambda_j \leftarrow \frac{3\lambda}{2(j\pi)^2}, j \leftarrow j + 1$
        $\widehat{\Delta}(F) \leftarrow EstSafety(\epsilon, \lambda_j, F)$
      **end if**
    **end if**
  **end while**
  return $F$

**Algorithm 2** *EstSafety* Subroutine

  Input: subset $F$
  Parameters: $\epsilon, \lambda_j \in (0,1)$
  $\widehat{\Delta}(F) \leftarrow 0$
  $\mathcal{T} \leftarrow$ sample $N_j = \frac{1}{2\epsilon^2} \ln \frac{2}{\lambda_j}$ trajectories
  **for** $\tau \in \mathcal{T}$ **do**
    **if** $\tau \not\subseteq F$ **then**
      $\widehat{\Delta}(F) \leftarrow \widehat{\Delta}(F) + \frac{1}{N_j}$
    **end if**
  **end for**
  return $\widehat{\Delta}(F)$

---

### 4.1 Algorithm Analysis

We define the event $\mathcal{E} = \{\forall i \; |\widehat{\Delta}(F_{i-1}) - \Delta(F_{i-1})| \leq \epsilon\}$, which states that all our *EstimateSafety* Subroutine estimations are accurate. We show that $\mathcal{E}$ holds with high probability using Hoeffding's inequality. In most of the analysis, we condition on $\mathcal{E}$ to hold.

The following theorem is the central component in the proof of the main theorem that follows it.

**Theorem 4.1.** *Given $\rho, \epsilon, \lambda \in (0,1)$,* FINDING SAFEZONE *Algorithm returns a subset $F \subseteq \mathcal{S}$ such that:*

1. *The escape probability is bounded from above by $\Delta(F) \leq 2\rho + 2\epsilon$, with probability $1 - \lambda$.*

2. *The expected size of $F$ given $\mathcal{E}$ is bounded by $\mathbb{E}[|F| \mid \mathcal{E}] \leq 2k^*$.*

3. *The sample complexity of the algorithm is bounded by $O\left(\frac{k^*}{\lambda\epsilon^2} \ln \frac{k^*}{\lambda} + \frac{Hk^*}{\rho\lambda}\right)$, and the running time is bounded by $O\left(\frac{Hk^*}{\lambda\epsilon^2} \ln \frac{k^*}{\lambda} + \frac{H^2k^*}{\rho\lambda}\right)$, with probability $1 - \lambda$.*

To obtain the main theorem, we run FINDING SAFEZONE Algorithm several times and return the smallest output set, $F$, see the next section for more details.

**Theorem 4.2.** *(main theorem) Given $\epsilon, \rho, \delta > 0$, if we run* FINDING SAFEZONE *for $\Theta(\frac{1}{\delta})$ times and return the smallest output set, $F \subseteq \mathcal{S}$, then with probability $\geq 0.99$*

1. *The escape probability is bounded by $\Delta(F) \leq 2\rho + 2\epsilon$.*

2. *The size of $F$ is bounded from above by $|F| \leq (2 + \delta)k^*$.*

3. *The total sample complexity and running time are bounded by $O(\frac{k^*}{\delta^2\epsilon^2} \ln \frac{k^*}{\delta} + \frac{Hk^*}{\rho\delta^2})$, and $O(\frac{Hk^*}{\delta^2\epsilon^2} \ln \frac{k^*}{\delta} + \frac{H^2k^*}{\rho\delta^2})$, respectively.*

Finding an almost $2\rho$-safe SAFEZONE , nearly $2k^*$ in size can be valuable. For example, it enhances the safety of popular commuting routes, promotes trust in autonomous vehicles, and aligns with potential regulatory restrictions, making most commutes driverless within this secure zone. Regarding sample complexity, we expect the dependency on $k^*$ to be nearly optimal. The reason why is the following. Consider an induced MC with $k^* - 1$ trajectories of size 2, each starting from an initial

state $s^0$, ends with a unique corresponding state $1, \ldots, k^* - 1$, and has a probability of $\frac{1-\rho}{k^*-1}$. If the MC has significantly more than $k^*$ states with non-zero probability, it would take at least $K^* - 1$ samples to find a $(\rho, k^*)$-SAFEZONE. As for the parameter $\delta$, this could be treated as a constant. For example, selecting $\delta = 1/3$ yields that we need to run the algorithm $6 \cdot \ln 300$ times and a solution of size $7/3k^*$ w.h.p.

## 4.2 Proof Technique

**Escape Probability Set Size Bounds.** To ease the presentation of the proof, we assume that $\widehat{\Delta}(F) = \Delta(F)$. For full proofs, we refer to Appendix C. This case is interesting on its own since if the policy and transition function are known, we can compute $\Delta(F)$ efficiently using dynamic programming (see Appendix E). As a result, event $\mathcal{E}$ always holds. In addition, it is clear that the termination of the algorithm implies that $\widehat{\Delta}(F) = \Delta(F) \leq 2\rho$, thus $F$ is $(2\rho + 2\epsilon)$-safe. The main challenge is bounding $|F|$.

A few notations before we start. Let $F^*$ denote a minimal $\rho$-safe set (of size $k^*$). Consider iteration $i$ inside the while-loop. The random variable $G_i(F)$ is the number of states in $F^*$ that are added to $F$ in iteration $i$ and $B_i(F)$ is the number of states added to $F$ in iteration $i$ that are not in $F^*$ ($G$ stands for *good* and $B$ for *bad*). For ease of presentation, from here on we write $G_i$ and $B_i$ instead of $G_i(F)$ and $B_i(F)$, respectively. Notice that the size of the output set is exactly $\sum_i B_i + G_i$ and that $\sum_i G_i \leq k^*$.

The main idea of the proof technique is to show that by adding trajectories according to the new distribution $Q_F$, we ensure that, in expectation, there are at least as many good states that are added to $F$ as bad states. Suppose the trajectory $\tau$ was chosen to be added to $F$ by the algorithm. If $\tau \subseteq F^*$, then $G_i$ is equal to $new_F(\tau)$ and $B_i = 0$. If $\tau \not\subseteq F^*$, then $B_i \leq new_F(\tau)$. Summarizing these observations, we have the following bounds

$$G_i \geq new_F(\tau) \cdot \mathbb{I}[\tau \subseteq F^*] \text{ and } B_i \leq new_F(\tau) \cdot \mathbb{I}[\tau \not\subseteq F^*],$$

where $\mathbb{I}[\cdot]$ is the indicator function.

Moreover, a direct consequence of the probability in which $\tau$ is added to $F$ is that for any set of trajectories $T$,

$$\mathbb{E}_{\tau \sim Q_F}[new_F(\tau) \cdot \mathbb{I}[\tau \in T]] = \sum_{\tau \in T} Q_F(\tau) new_F(\tau)$$

$$= \frac{1}{Z} \sum_{\tau \in T, new_F(\tau) \neq 0} \left( \frac{\Pr[\tau]}{new_F(\tau)} \right) new_F(\tau) \qquad (1)$$

$$= \frac{1}{Z} \Pr_\tau[\tau \in T \land new_F(\tau) \neq 0],$$

where $Z$ is the normalization factor of $Q_F$.

To bound the size of $F$, we want to show that the algorithm does not add too many states outside of $F^*$. We therefore bound $\mathbb{E}[B_i]/\mathbb{E}[G_i]$, where the expectations are over the trajectory $\tau$ that is added to $F$ according to $Q_F$. Applying Equation (1) twice, once with $T = \{\tau \mid \tau \subseteq F^*\}$ and once with $T = \{\tau \mid \tau \not\subseteq F^*\}$, we bound the ratio between $B_i$ and $G_i$ by

$$\frac{\mathbb{E}[B_i]}{\mathbb{E}[G_i]} \leq \frac{\Pr_\tau[\tau \not\subseteq F^* \land new_F(\tau) \neq 0]}{\Pr_\tau[\tau \subseteq F^* \land new_F(\tau) \neq 0]}. \qquad (2)$$

We know that $\Pr_\tau[\tau \not\subseteq F^*]$ is always smaller than $\rho$, so the numerator is $\leq \rho$. A lower bound for the denominator is $\Pr_\tau[new_F(\tau) \neq 0] - \Pr_\tau[\tau \not\subseteq F^*]$. In addition, whenever the algorithm is inside the main loop, the safety is at least $\Pr_\tau[new_F(\tau) \neq 0] = \Delta(F) > 2\rho$. Thus, the denominator is at least $\rho$. Hence, the RHS of (2) is less or equal to 1, thus

$$\mathbb{E}[B_i] \leq \mathbb{E}[G_i]. \qquad (3)$$

This completes the proof because we know that the algorithm does not add too many states outside of $F^*$. More precisely,

$$\mathbb{E}[|F|] = \mathbb{E}\left[ \sum_i B_i + G_i \right] \leq \mathbb{E}\left[ 2 \sum_i G_i \right] \leq 2k^*.$$

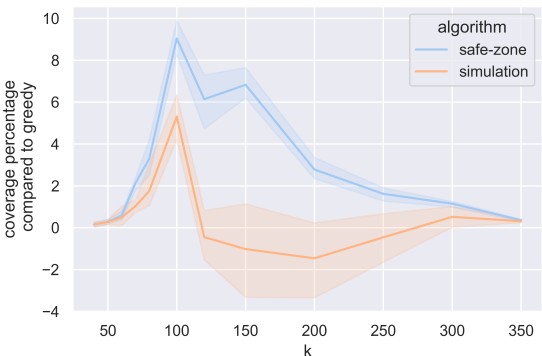

Figure 1: Coverage percentage: difference from GREEDY Algorithm.

**Sample Complexity.** To discuss the sample complexity, we drop the assumption that the MC is known to the learner and use *EstimateSafety* Subroutine to approximate $\Delta(F)$. The number of calls to *EstimateSafety* is bounded by the size of the output set, $F$. Hence, this part of the sample complexity is bounded by $|F| \cdot N_{|F|}$ and we show that is $O(\frac{k^*}{\epsilon^2} \log k^*)$. Another source of sampling is trajectories sampled for purposes of potentially adding them to $F$. Observe that at any iteration the set $F$ has an escape probability of at least $2\rho$, and each trajectory that escapes $F$ is accepted with a probability of at least $1/H$. This implies a lower bound for the probability that a random trajectory is accepted is $2\rho/H$. This gives an upper bound of $\frac{2|F|\rho}{H}$ for the expected sample complexity.

**Amplification.** Theorem 4.1 shows that if $\mathcal{E}$ holds, then the set size, $|F|$, is bounded *in expectation* by $2k^*$. As $\Pr[\mathcal{E}] \geq 1 - \lambda$ implies, from Markov's inequality, that the size $(2 + \delta)k^*$ with small probability of about $\delta + \lambda = O(\delta)$. If we want to make sure that the actual size is at most $(2 + \delta)k^*$ with high probability, we can repeat the process about $\Theta\left(\frac{1}{\delta}\right)$ times and take the smallest size set.

## 5 Empirical Demonstration

This section demonstrates the qualitative and quantitative performance of the described algorithms in the paper. For additional figures, we refer the reader to Appendix D.

**The MDP.** We focus on a grid of size $N \times N$, for some parameter $N$. The agent starts off at mid-left state, $(0, \lfloor \frac{N}{2} \rfloor)$ and wishes to reach the (absorbing) goal state at $(N-1, \lfloor \frac{N}{2} \rfloor)$ with a minimal number of steps. At each step, it can take one of four actions: up, down, right, and left by 1 grid square. With probability $0.9$, the intended action is performed and with probability $0.1$ there is a drift down. The agent stops either way after $H = 300$ steps.

### 5.1 FINDING SAFEZONE vs. naive approaches

To compare the FINDING SAFEZONE Algorithm to the naive approaches presented in Section 3 we focus on the policy that first goes to the right and when it reaches the rightmost column, it goes up. The policy is described in the appendix, Figure 6(d). We take $N = 30$ and 2000 episodes.

We run the FINDING SAFEZONE , GREEDY, and SIMULATION algorithms, and estimate their coverage based on a test set containing 2000 random trajectories. Figure 1 depicts the trajectories coverage of each algorithm minus the coverage of the GREEDY algorithm. For a figure with absolute values, we refer the reader to Figure 6(b) in the appendix. We see that the new algorithm exhibits better performance compared to its competitors. We also see that taking less than $30\%$ of the states ($k = 250$ out of 900 states) is enough to get coverage of more than $80\%$ of the trajectories.

Figures 5(a),5(b) show the sets found for $k = 60$ both by the *Finding* SAFEZONE Algorithm and GREEDY. We see that GREEDY chooses an unconnected set for this small $k$, leading to a coverage of $0$. While the new algorithm chooses a few states which consist of several trajectories, thus leading to a coverage larger than $0$.

# 6    Discussion and Open Problems

In this paper, we have introduced the SAFEZONE problem. We have shown that it is NP-hard, even when the model is known, and designed a nearly $(2\rho, 2k^*)$ approximation algorithm for the case where the model and policy are unknown to the algorithm. Beyond improving the approximation factors (or showing that it cannot be done unless $P = NP$), a natural direction for future work is the following. Given a small $\rho > 0$ and a (known or unknown to the learner) MDP, find a policy with a small $\rho-$safe subset. If the value of the policy, when restricted to the SAFEZONE states, is close to the optimal value of the original MDP, restricting the policy to the SAFEZONE states generates a compact policy representation with a value close to optimal, and most trajectories are completed in the SAFEZONE.

## Acknowledgement

This project has received funding from the European Research Council (ERC) under the European Union's Horizon 2020 research and innovation program (grant agreement No. 882396), by the Israel Science Foundation (grant number 993/17), Tel Aviv University Center for AI and Data Science (TAD), and the Yandex Initiative for Machine Learning at Tel Aviv University.

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

## Supplementary Material

## A   Hardness

In this section, we show that SAFEZONE is NP-hard to solve, and this is why approximation is necessary. Moreover, SAFEZONE is hard even if the MC and the optimal $\rho-$safe size, $k^*$ are known. Our starting point is the NP-hardness of regular cliques. The REGULARCLIQUE$(G, k_c)$ problem gets as an input (i) a regular graph $G$ with $n$ nodes where each node has degree $d$, and (ii) an integer $k_c$. It returns whether $G$ contains a clique of size $k_c$. Whenever $G$ and $k_c$ are clear from the context we simply write REGULARCLIQUE. The following fact follows, e.g., from Brandes et al. (2016).

**Fact A.1.** REGULARCLIQUE *is NP-hard.*

**Markov chain (random walk).** Fix a graph $G = (V, E)$ and a starting vertex $v_0 \in V$. The graph induces a Markov Chain (random walk) in the following way. The states of the process correspond to the vertices $V$ in the graph $G$. The transition function is defined as $P(v|u) = \frac{1}{d} \cdot \mathbb{1}_{(u,v) \in E}$, where $d$ is the degree of any node. The process starts from node $v_0$ and then proceeds according to the transition function $P$ for $H$ steps.

**Reduction.** To prove the hardness of SAFEZONE , we show how to solve REGULARCLIQUE given a solver to SAFEZONE. For each vertex $v \in V$, run an algorithm for SAFEZONE  with horizon $H = 2$, $k = k_c$, and $\rho = 1 - \left(\frac{k_c-1}{d}\right)^2$, and $v$ as the starting state. If there is at least one run of the algorithm that returns YES, then the final answer is YES. Otherwise, the answer is NO. Note that this reduction is efficient.

**Theorem A.2.** *For every graph $G = (V, E)$ and an integer $k_c$ there exists a clique of size $k_c$ in $G$ $\iff$ there exists $v \in V$ such that* SAFEZONE$(V, v_0 = v, P, k_c, \rho)$ *returns YES.*

*Proof.* ($\implies$) If there is a clique of size $k_c$, then we can take the corresponding $k$ states. The probability to remain in this subset is at least $\left(\frac{k-1}{d}\right)^2$ (remember that $H = 2$). Thus, an exact solver for SAFEZONE must return YES.

($\impliedby$) Suppose there is no clique of size $k$. Assume by contradiction that the reduction (algorithm) returns YES. Let $s_0$ be a vertex which was the starting state from the running instance which the YES came from and let $\hat{F}$ denote the output of SAFEZONE . We will show that the probability to remain in any subset of size $k$ is smaller than $\left(\frac{k-1}{d}\right)^2$.

Since there is no clique of size $k$ in $G$, we know that $\hat{F}$ is not a clique. It therefore follows that there exists at least two vertexes, $s_a, s_b \in V$ such that $(s_a, s_b) \notin E$.

We will now bound the probability of escape from state $s_0$ by exhaustion.

1. If $s_0 \neq s_a$, then

$$\Pr[escape\ from\ s_0] \geq \Pr[t = 1 : (s_0, s'), s' \notin \hat{F}]$$

$$+ \Pr[t = 1 : (s_0, s), s \neq s_a] \cdot \Pr[t = 2 : (s, s'), s' \notin \hat{F}|t = 1 : (s_0, s), s \neq s_a]$$

$$+ \Pr[t = 1 : (s_0, s_a)] \cdot \Pr[t = 2 : (s_a, s'), s' \notin \hat{F}|t = 1 : (s_0, s_a)]$$

$$= \frac{d - (k - 1)}{d} + \frac{k - 2}{d} \cdot \frac{d - (k - 1)}{d} + \frac{1}{d} \cdot \frac{d - (k - 2)}{d}$$

$$= 1 - \frac{k - 1}{d} + \frac{k - 2}{d} - \frac{(k - 2)(k - 1)}{d^2} + \frac{1}{d} - \frac{k - 2}{d^2} =$$

$$1 - \frac{k - 2}{d^2}(k - 1 + 1) = 1 - \frac{k(k - 2)}{d^2}$$

Hence

$$\Pr[staying] \leq \frac{k(k - 2)}{d^2} < \frac{(k - 1)^2}{d^2}.$$

2. If $s_0 = s_a$, then

$$\Pr[escape\ from\ s_0] \geq \Pr[t = 1 : (s_0, s'), s' \notin \hat{F}]$$

$$+ \Pr[t = 1 : (s_0, s), s \in \hat{F}] \cdot \Pr[t = 2 : (s, s'), s' \notin \hat{F}|t = 1 : (s_0, s), s \in \hat{F}]$$

$$= \frac{d - (k - 2)}{d} + \frac{k - 2}{d} \cdot \frac{d - (k - 1)}{d}$$

$$= 1 - \frac{k - 2}{d} + \frac{k - 2}{d} - \frac{(k - 2)(k - 1)}{d^2}$$

$$= 1 - \frac{(k - 2)(k - 1)}{d^2}$$

Hence

$$\Pr[staying] \leq \frac{(k - 2)(k - 1)}{d^2} < \frac{(k - 1)^2}{d^2}.$$

$\square$

Given an environment, a policy, and a SAFEZONE , one could compute exactly how much safe it is (see Appendix E for details), from which we deduce our following corollary.

**Corollary A.3.** SAFEZONE *is NP-complete.*

We note that for $H = 1$, the GREEDY AT EACH STEP Algorithm is optimal.

# B Proofs of Section 3

## B.1 Greedy by Threshold Algorithm

A naive approach to the SAFEZONE problem is to return all states $s \in \mathcal{S}$ with probability $p(s) \geq \beta$, for some parameter $\beta > 0$, see Algorithm 3.

---
**Algorithm 3** Greedy by Threshold

---
Parameter: $\beta > 0, \{p(s)\}_{s \in \mathcal{S}}$
return $\{s \in \mathcal{S} : p(s) \geq \beta\}$

---

**Lemma 3.1.** *For any $\rho, \beta \in (0, 1)$, the* GREEDY BY THRESHOLD ALGORITHM *returns a set that is* $\left(\rho + k^*\beta, \frac{H}{\beta}\right) -$ SAFEZONE. *In particular, for $\beta = \frac{\rho}{k^*}$, this set is* $\left(2\rho, \frac{k^*H}{\rho}\right) -$ SAFEZONE.

*Proof.* There are at most $\frac{H}{\beta}$ states with probability $p(s) \geq \beta$. Thus $|F| \leq \frac{H}{\beta}$.

Denote by $F^*$ an optimal $(\rho, k^*) -$ SAFEZONE set. By the law of total probability,

$$\Pr_\tau[\tau \not\subseteq F] \leq \Pr_\tau[\tau \not\subseteq F^*] + \Pr_\tau[\tau \subseteq F^* \setminus F].$$

Looking at the R.H.S of the inequality, the left term is smaller than $\rho$ by the definition of SAFEZONE. The right term is equal to the probability of reaching a state in $F^*$ that its probability is smaller than $\beta$, i.e., a state in $F^* \setminus F$.

Using union bound, this can be bounded by $k^*\beta$. $\square$

**Lemma B.1.** *For every $\rho \in (0, 1/2), H \in \mathbb{N}$, there exists an MDP and a minimal integer $k$ such that the MDP has a $(\rho, k) -$ SAFEZONE , but for $\beta = \rho/k$ GREEDY BY THRESHOLD Algorithm returns $F$ with escape probability $\leq 2\rho$ and of size $|F| = \Omega(H/\beta)$.*

*Proof.* Fix $\rho \in (0, 1)$. For ease of the presentation, we will assume that $\frac{1-\rho}{\beta}$ is an integer (if not, it should be rounded to the nearest integer). Define $A$ to contain $\frac{1-\rho}{\beta} \cdot H$ states, $B$ to contain $k - 1$ states, and $\mathcal{S} = \{s_0\} \cup A \cup B$. Consider the following MDP with states $\mathcal{S}$ and starting state $s_0$. The transition function is defined as follows:

- For every $i \in A$, $\Pr[s^A_{1,i}|s_0] = \beta$ and for every $j \in [H-1]$, $\Pr[s^A_{j+1,i}|s^A_{j,i}] = 1$.

- For $s \in B$, $\Pr[s|s_0] = \frac{1-\rho}{k-1}$

- For $s \in B$, $\Pr[s|s] = 1$

The MDP is illustrated in Figure 2. Clearly, $\{s_0\} \cup B$ is a $(\rho, k)-$SAFEZONE . In addition, GREEDY BY THRESHOLD ALGORITHM returns the set of all states, as for every state $s \in A$ we have that $p(s) = \beta$, $p(s_0) = 1 > \rho \geq \beta$, and for every $s \in B$ we have that $p(s) = \frac{1-\rho}{k-1} > \frac{\rho}{k} = \beta$. Thus the size of the returned set is $\mathcal{S}$ is $\Omega(H/\beta)$, which completes the proof. $\qquad\square$

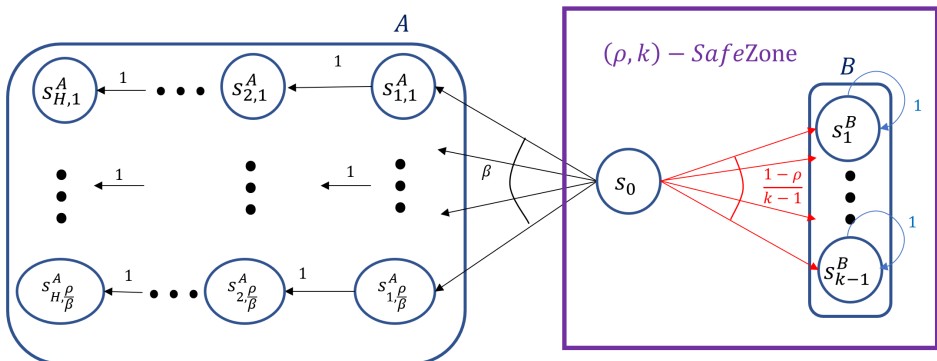

Figure 2: Lower bound for GREEDY BY THRESHOLD Algorithm.

## B.2 Simulation Algorithm

---
**Algorithm 4** Simulation Algorithm
---
Input: $m = \frac{1}{\beta} \ln \frac{k^*}{0.005}$
$F \leftarrow \{s_0\}$
**for** $i = 1 \ldots m$ **do**
   $\tau \leftarrow$ choose a random trajectory
   $F \leftarrow F \cup \tau$
**end for**
return $F$

---

**Lemma 3.2.** *Fix $\rho, \beta \in (0,1)$. With probability at least $0.99$, SIMULATION Algorithm returns a set that is $\left(\rho + k^*\beta, O(k^* + \frac{\rho H \ln k^*}{\beta})\right)$ − SAFEZONE. In particular, for $\beta = \frac{\rho}{k^*}$, this set is $(2\rho, O(k^* H \ln k^*))$ − SAFEZONE.*

*Proof.* Denote by $F^*$ the optimal $(\rho, k^*)$ − SAFEZONE set. By the law of total expectation, we can split $\mathbb{E}[|F|]$ into two parts, depending on whether trajectories are entirely in $F^*$ or not:

- Trajectories that are entirely in $F^*$ contribute at most $k^*$ states to $F$.

- A trajectory that is not contained in $F^*$ contributes at most $H$ states to $F$.

Thus,

$$\mathbb{E}[|F|] \leq k^* + \rho \cdot \left(\frac{1}{\beta} \ln \frac{k^*}{0.005}\right) \cdot H = O\left(k^* + \frac{\rho H \ln k^*}{\beta}\right).$$

We use Markov's inequality to get the desired bound on $|F|$.

For the safety, we first denote the set of all states in $F^*$ with probability at least $\beta$ as $\Gamma = \{s \in F^* \mid p(s) \geq \beta\}$. We will show that with probability at least $0.9995$, it holds that $\Gamma \subseteq F$, which will prove our claim, similarly to Lemma 3.1.

For a fixed state $s \in \Gamma$, the probability that $s \notin F$ is bounded by $(1 - p(s))^{\frac{1}{\beta} \ln \frac{k^*}{0.005}} \leq e^{-\frac{\beta}{\beta} \cdot \ln \frac{k^*}{0.005}} = \frac{0.005}{k^*}$. Using union bound, the probability that there is a state $s \in \Gamma$ which is not in $F$ is bounded by $k^* \cdot \frac{0.005}{k^*} = 0.005$.

In other words, with probability at least $0.995$, $\Gamma \subseteq F$, thus implementing the greedy approach in Algorithm 3 and proving that the probability that a random trajectory escapes $F$ is bounded by $\rho + k^* \beta$. $\qquad\square$

**Lemma B.2.** *For every $\rho, \gamma \in (0, 1)$, $H, k \in \mathbb{N}$, and $\beta = \frac{\rho}{k}$, there is an integer $r \in \mathbb{N}$ and MDP with $(\rho, k)-$SAFEZONE, but with probability $\geq 1 - \gamma$, SIMULATION algorithm returns $F$ of size $\mathbb{E}[|F|] \geq kH \ln k$ with escape probability $\Delta(F) = O(\rho)$.*

*Proof.* Fix $\rho, \gamma \in (0, 1)$. Recall that $m = \frac{1}{\beta} \ln \frac{k^*}{0.005}$ and take $r = \lceil \frac{m^2}{\gamma} \rceil$. Define $A$ to contain $rH$ states, $B$ to contain $k - 1$ states, and $\mathcal{S} = \{s_0\} \cup A \cup B$.

Consider the following MDP with states $\mathcal{S}$ and starting state $s_0$. The transition function is defined as follows:

- For every $i \in A$, $\Pr[s_{1,i}^A | s_0] = \frac{\rho}{r}$ and for every $j \in [H - 1]$, $\Pr[s_{j+1,i}^A | s_{j,i}^A] = 1$.

- For $s \in B$, $\Pr[s | s_0] = \frac{1 - \rho}{k - 1}$

- For $s \in B$, $\Pr[s | s] = 1$

The MDP is illustrated in Figure 3.

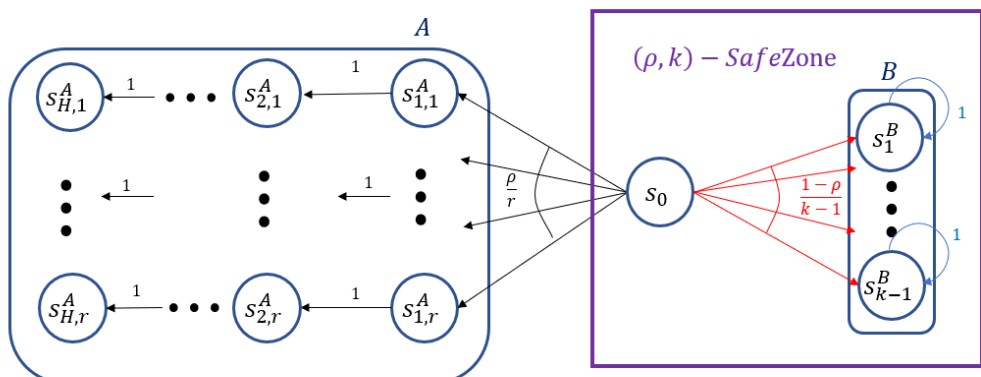

Figure 3: Lower bound for SIMULATION Algorithm.

The set $B \cup \{s_0\}$ is $\rho-$safe with $k$ states.

We will show that:

- After adding $\geq \frac{1}{\beta} \ln k = \frac{k}{\rho} \ln k$ random trajectories, with probability $\geq 1 - \gamma$ we have that $|F| \geq kH \ln k$.

- After adding $m$ random trajectories, we have that with high probability $F^* \subseteq F$, thus $\Delta(F) \leq \Omega(\rho)$.

To prove the first property, we claim that with probability $\geq 1 - \gamma$, every time we add a trajectory $\tau$ such that $\tau \cap A \neq \emptyset$, we add $H$ new states.

Notice that if we ignore $s_0$, trajectories in $A$ are entirely unconnected, and each trajectory is chosen randomly with probability $\Pr[s_{1,i}^A | s_0] = \frac{\rho}{r}$. This yields that if $s_{1,i}^A \notin F$, then $s_{j,i}^A \notin F$ for every $j \in [H]$. As a result, every time we add a new $s_{1,i}^A$ to $F$, we add $H - 1$ more states to $F$. Let $N$ denote the number of trajectories sampled with states from $A$. The probability that their intersection contains only $s_0$ is

$$\frac{r \cdot (r-1) \cdot \ldots \cdot (r - N)}{r^N} \geq \left(\frac{r - N}{r}\right)^N = \left(1 - \frac{N}{r}\right)^N \geq 1 - \frac{N^2}{r} = 1 - \gamma.$$

From the structure of the MDP, we have that $\mathbb{E}[N] = \rho m$. Therefore, with probability $\geq 1 - \gamma$,

$$\mathbb{E}[|F|] \geq \mathbb{E}[N] \cdot H = \rho \cdot m \cdot H \geq \rho \cdot \frac{1}{\beta} \ln k \cdot H = kH \ln k.$$

The second property follows from Lemma 3.2. $\qquad\square$

## B.3 Greedy at Each Step

---
**Algorithm 5** Greedy at Each Step

---
Input: $\rho > 0, \{p(s)\}_{s \in \mathcal{S}}$
$F \leftarrow \{s_0\}$
**for** $i = 1 \ldots H$ **do**
  Sort states in $\mathcal{S}_i$, $p(s_i^1) \geq \ldots \geq p(s_i^{|\mathcal{S}_i|})$
  $j^* \leftarrow \arg\min_{j \in [|\mathcal{S}_i|]} \sum_{r=1}^{j} p(s_i^r) \geq 1 - \rho$
  $F \leftarrow F \cup \left\{s_i^1, \ldots s_i^{j^*}\right\}$
**end for**
return $F$

---

**Lemma 3.3.** *For any $\rho \in (0,1)$, if the MDP is layered,* GREEDY AT EACH STEP ALGORITHM *returns a set that is $(\rho H, k^*) -$ SAFEZONE.*

*Proof.* Take a random trajectory $\tau = (s_1, s_2, \ldots)$. For every $s_i \in \tau$, the probability that $s_i \notin F$ is bounded by $\rho$, thus using union bound, the probability that $\tau$ has state $s_i$ such that $s_i \notin F$ is at most $\rho H$.

The construction of $F$ guarantees that $F$ is the minimal subset of states such that for every $i$, the probability that $s_i$ is in the subset is at least $1 - \rho$. Assume by contradiction that $|F| > k^*$. Then there is a time step $i$ such that $\Pr[s_i \in F^*] < 1 - \rho$, which is a contradiction, since $\Pr[\tau \in F^*] \leq \min_i \Pr[s_i \in F^*]$.

$\qquad\square$

**Lemma B.3.** *For any $\rho \in (0,1)$, there is an MDP and an integer $k$ such that there is a $(\rho, k) -$ SAFEZONE, but* GREEDY AT EACH STEP *Algorithm returns $F$ with escape probability $\Delta(F) \geq \Omega(H\rho)$.*

*Proof.* Fix $\rho \in (0,1)$ and take $k = 3H + 1$.

Consider the MDP illustrated in Figure 4. The set $\{s_0\} \cup \{s_1^i\}_i \cup \{s_2^i\}_i \cup \{s_3^i\}_i$ form a $(\rho, 3H + 1) -$ SAFEZONE .

We will prove by induction that the for every time $i$,

- $p(s_1^i) = 1 - 2\rho$,

- $p(s_2^i) = p(s_3^i) = p(s_4^i) = \frac{\rho}{2}$, and

- For every $j \in \{5, \ldots, k + 4\}$, $p(s_j^i) = \frac{\rho}{2k}$.

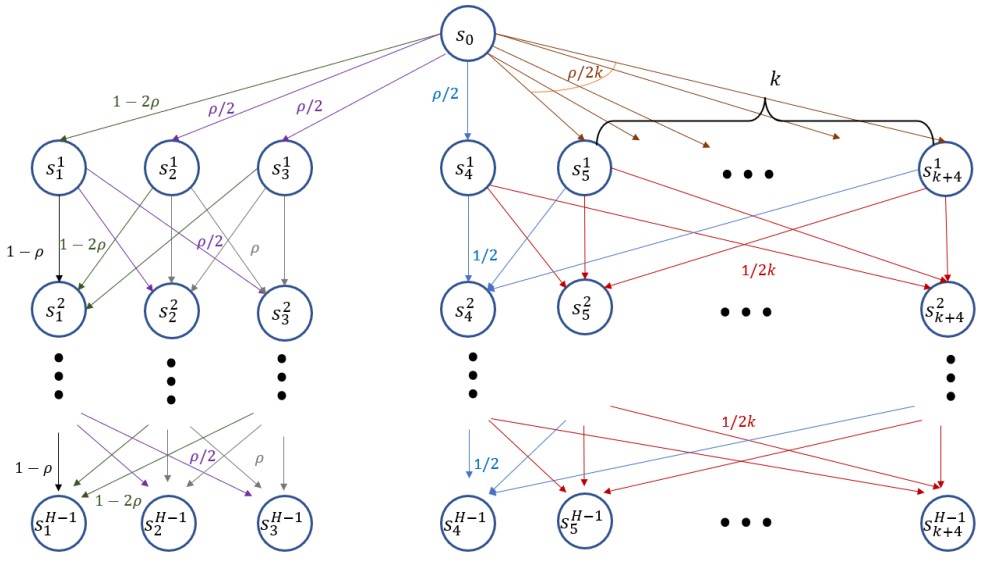

Figure 4: Lower bound for GREEDY AT EACH STEP Algorithm.

It is easy to see that the two properties hold for $i = 1$.

For $i > 1$,

$$p(s_1^i) = p(s_1^{i-1})(1 - \rho) + p(s_2^{i-1})\frac{\rho}{2} + p(s_3^{i-1})\frac{\rho}{2} = (1 - 2\rho)(1 - \rho) + 2(1 - 2\rho)\frac{\rho}{2} = 1 - 2\rho$$

$$p(s_2^i) = p_{i-1}(s_1^{i-1})\frac{\rho}{2} + p(s_2^{i-1})\rho + p(s_3^{i-1})\rho = (1 - 2\rho)\frac{\rho}{2} + \frac{\rho^2}{2} + \frac{\rho^2}{2} = \frac{\rho}{2}$$

Similarly, $p(s_3^i) = \frac{\rho}{2}$.

$$p(s_4^i) = \frac{1}{2}p(s_4^{i-1}) + \sum_{j=5}^{k+4} \frac{p(s_j^{i-1})}{2} = \frac{\rho}{4} + k\frac{\rho}{4k} = \frac{\rho}{2}$$

For every $j \in \{5, \ldots, k+4\}$,

$$p(s_j^i) = \frac{1}{2k}p(s_4^{i-1}) + \sum_{m=5}^{k+4} \frac{p(s_m^{i-1})}{2k} = \frac{\rho}{4k} + k\frac{\rho}{4k^2} = \frac{\rho}{2k}.$$

The algorithm might return $\{s_0\} \cup \{s_1^i\}_i \cup \{s_2^i\}_i \cup \{s_4^i\}_i$, i.e., instead of taking $\cup_i\{s_3^i\}_i$ it takes $\cup_i\{s_4^i\}_i$. Finally, the observation $\Delta(\{s_0\} \cup \{s_1^i\}_i \cup \{s_2^i\}_i \cup \{s_4^i\}_i) \geq \frac{\rho H}{4}$ completes the proof. $\square$

## C   Proofs of Section 4

For convenience, we state here Hoeffding's inequality.

**Lemma C.1.** *[Hoeffding's Inequality] Let $y_1, \ldots, y_N$ be independent random variables such that $y_i \in [a, b]$ for every $y_i$ with probability 1. Then, for any $\epsilon > 0$,*

$$\Pr\left[\left|\frac{1}{N}\sum_{i=1}^{N} y_i - \mathbb{E}[y_i]\right| \geq \epsilon\right] \leq 2e^{-2N\epsilon^2/(b-a)^2}.$$

## C.1 Proof of Theorem 4.1

In this section, we provide a complete proof for Theorem 4.1. Throughout the section, we define a few terms and notions. We will start with proving guarantees regarding a single iteration of the while–loop.

Recall that $F^*$ denotes a minimal $\rho-$safe set (of size $k^*$). If there are multiple optimal solutions, choose one arbitrarily. For the convince of analysis, we denote the values of the algorithm variables at the end of each iteration $i$ of the while–loop by $\tau_i, F_i, accept_i$. Let $j(i)$ denote the value of variable $j$ during the $i-$th call to *EstimateSafety* Subroutine. In addition, let $N_i$ denote the number of trajectories sampled for the $j-$th time of calling Subroutine *EstimateSafety*, i.e., $N_i = \frac{1}{2\epsilon^2} \ln \frac{2}{\lambda_{j(i)}}$ for $j(i) \leq i$.

For ease of presentation, we recall some of the definitions from the proof technique description. We say that a trajectory $\tau$ is *good* if all the states in $\tau$ are in $F^*$ and *bad* if it escapes it. I.e., a trajectory is good if $\tau \subseteq F^*$ and bad if $\tau \not\subseteq F^*$. Additionally, we say that a state $s \in \mathcal{S}$ is *good* if it is in $F^*$ and *bad* otherwise. Namely, a state $s$ is good if $s \in F^*$ and bad if $s \notin F^*$. Let $G_i(F_{i-1})$ and $B_i(F_{i-1})$ be the number of good and bad states added to $F_{i-1}$ in iteration $i$, respectively (notice that $G_i(F_{i-1})$ and $B_i(F_{i-1})$ are random variables that depends on $F_{i-1}$). For short, whenever it is clear from the context, we write $G_i$ and $B_i$ respectively.

The following lemma bounds the error in approximating the escape probability.

**Lemma C.2.** *Let $F_{i-1} \subseteq \mathcal{S}$ be a subset of of states and $\epsilon, \lambda_j > 0$ be some parameters. Let $S_i$ be a sample of $N_i \geq \frac{1}{2\epsilon^2} \ln \frac{2}{\lambda_{j(i)}}$ i.i.d. random trajectories. Then,*

$$\Pr_{S_i} \left[ \left| \widehat{\Delta}(F_{i-1}) - \Delta(F_{i-1}) \right| \geq \epsilon \right] \leq \lambda_j.$$

*Also, as $\lambda_j = \frac{3\lambda}{2(\pi j)^2}$,*

$$\Pr \left[ \exists i \ \left| \widehat{\Delta}(F_{i-1}) - \Delta(F_{i-1}) \right| \geq \epsilon \right] \leq \lambda/4,$$

*Where the last probability is over all the samples $S_i$ made by EstimateSafety Subroutine.*

*Proof.* The first part follows directly from Hoeffding's inequality by taking $y_i = \mathbb{I}[\tau \not\subseteq F]$.

Assigning $\lambda_j = \frac{3\lambda}{2(\pi j)^2}$ and applying union bound, we get

$$\Pr \left[ \exists i \ \left| \widehat{\Delta}(F_{i-1}) - \Delta(F_{i-1}) \right| \geq \epsilon \right] \leq \sum_i \Pr_{S_i} \left[ \left| \widehat{\Delta}(F_{i-1}) - \Delta(F_{i-1}) \right| \geq \epsilon \right]$$

$$\leq_{(*)} \sum_{j(i)} \lambda_{j(i)} \leq \sum_{j=1}^{\infty} \lambda_j = \sum_{j=1}^{\infty} \frac{3\lambda}{2(\pi j)^2} = \frac{\lambda}{4}.$$

The inequality marked by $(*)$ follows from the fact that $\Delta(F)$ is estimated once for every time $j$ increases. $\qquad\square$

We define the event that *EstimateSafety* always provides good estimations by

$$\mathcal{E} = \{ \forall i \ \left| \widehat{\Delta}(F_{i-1}) - \Delta(F_{i-1}) \right| \leq \epsilon \}.$$

By the above, we have that $\Pr[\mathcal{E}] \geq 1 - \lambda/4$.

In the following lemma we assume that if the current escape probability is at least $2\rho$, then the fraction of bad trajectories that escape $F_{i-1}$ is bounded from above by the fraction of good trajectories that escape $F_{i-1}$.

**Lemma C.3.** *Let $\rho > 0$ and assume that $\Delta(F_{i-1}) \geq 2\rho$. Then,*

$$\Pr_{\tau}[new_{F_{i-1}}(\tau) \neq 0 \wedge \tau \not\subseteq F^*] \leq \Pr_{\tau}[new_{F_{i-1}}(\tau) \neq 0 \wedge \tau \subseteq F^*],$$

*where the probabilities are over random trajectories.*

*Proof.* To prove the lemma, we will bound the probability $\Pr_\tau[new_{F_{i-1}}(\tau) \neq 0 \wedge \tau \not\subseteq F^*]$ from above and the probability $\Pr_\tau[new_{F_{i-1}}(\tau) \neq 0 \wedge \tau \subseteq F^*]$ from below. Since $\Delta(F^*) \leq \rho$,

$$\Pr_\tau[new_{F_{i-1}}(\tau) \neq 0 \wedge \tau \not\subseteq F^*] \leq \Pr_\tau[\tau \not\subseteq F^*] \leq \rho. \tag{4}$$

The assumption $\Delta(F_{i-1}) \geq 2\rho$ implies that

$$2\rho \leq \Delta(F_{i-1}) = \Pr_\tau[new_{F_{i-1}}(\tau) \neq 0] = \Pr_\tau[new_{F_{i-1}}(\tau) \neq 0 \wedge \tau \subseteq F^*] + \Pr_\tau[new_{F_{i-1}}(\tau) \neq 0 \wedge \tau \not\subseteq F^*]$$

$$\leq \Pr_\tau[new_{F_{i-1}}(\tau) \neq 0 \wedge \tau \subseteq F^*] + \Pr_\tau[\tau \not\subseteq F^*] \leq \Pr_\tau[new_{F_{i-1}}(\tau) \neq 0 \wedge \tau \subseteq F^*] + \rho,$$

hence

$$\rho \leq \Pr_\tau[new_{F_{i-1}}(\tau) \neq 0 \wedge \tau \subseteq F^*]. \tag{5}$$

Putting (4) and (5) together yields the statement. $\square$

Now, as long as the algorithm is inside the while–loop (i.e., the escape probability holds $\widehat{\Delta}(F) > 2\rho + \epsilon$), it follows that $\Delta(F) \geq 2\rho$ with high probability from Lemma C.2. Combining it with Lemma C.3 would yield that with high probability over a random trajectory, if the trajectory escapes $F$ then in expectation, it is at least as likely to be good as it is to be bad.

We move on to show the main ingredient of the proof, namely that for any iteration, with high probability, the expected number of good states added to the current set $F$ is larger or equal to the expected number of bad states.

For every iteration $i$ in which we sample $\tau_i$ both $G_i$ and $B_i$ depends on the following:

1. The realizations of the sampled trajectory, $\tau_i$, and in particular on $new_{F_{i-1}}(\tau_i)$.
2. The probability of adding it to $F$, i.e., $1/new_{F_{i-1}}(\tau_i)$.

Next, we prove Equation (3).

**Lemma C.4.** *Assume event $\mathcal{E}$ holds. Thus, for all iterations $i$ inside the while–loop we have*

$$\mathbb{E}[B_i|F_{i-1}] \leq \mathbb{E}[G_i|F_{i-1}],$$

*where the expectation is over the trajectory $\tau$ that is sampled from the MC dynamics and added to $F_{i-1}$ according to $Q_{F_{i-1}}$.*

*Proof.* Since event $\mathcal{E}$ holds, we have that $\Delta(F_{i-1}) \geq 2\rho$ as long as we do not terminate in iteration $i$.
We can use it to bound $\mathbb{E}_\tau[B_i|F_{i-1}]$ by

$$\begin{aligned}
\mathbb{E}_\tau[B_i|F_{i-1}] &\leq \sum_{h=1}^{H} \frac{\Pr_\tau[new_{F_{i-1}}(\tau) = h \wedge \tau \not\subseteq F^*]}{h} \cdot h \\
&= \Pr_\tau[new_{F_{i-1}}(\tau) \neq 0 \wedge \tau \not\subseteq F^*] \underbrace{\leq}_{Lemma\ C.3} \Pr_\tau[new_{F_{i-1}}(\tau) \neq 0 \wedge \tau \subseteq F^*] \\
&= \sum_{h=1}^{H} \frac{\Pr_\tau[new_{F_{i-1}}(\tau) = h \wedge \tau \subseteq F^*]}{h} \cdot h \leq \mathbb{E}_\tau[G_i|F_{i-1}].
\end{aligned}$$

$\square$

**Theorem 4.1.** *Given $\rho, \epsilon, \lambda \in (0,1)$, FINDING SAFEZONE Algorithm returns a subset $F \subseteq \mathcal{S}$ such that:*

1. *The escape probability is bounded from above by $\Delta(F) \leq 2\rho + 2\epsilon$, with probability $1 - \lambda$.*

2. *The expected size of $F$ given $\mathcal{E}$ is bounded by $\mathbb{E}[|F| \mid \mathcal{E}] \leq 2k^*$.*

3. *The sample complexity of the algorithm is bounded by $O\left(\frac{k^*}{\lambda\epsilon^2} \ln \frac{k^*}{\lambda} + \frac{Hk^*}{\rho\lambda}\right)$, and the running time is bounded by $O\left(\frac{Hk^*}{\lambda\epsilon^2} \ln \frac{k^*}{\lambda} + \frac{H^2 k^*}{\rho\lambda}\right)$, with probability $1 - \lambda$.*

*Proof.* Assume that the event $\mathcal{E}$ holds, and recall that

$$\Pr[\mathcal{E}] \geq 1 - \lambda/4. \tag{6}$$

We start with the first clause. Since the event $\mathcal{E}$ holds, Lemma C.2 in particular implies that $\Delta(F) \leq 2\rho + 2\epsilon$, hence the first clause holds. For second clause, we will bound $\mathbb{E}[|F| \mid \mathcal{E}]$ from above by $2k^*$. Since $\mathcal{E}$ holds, we have that $\Delta(F_{i-1}) \geq 2\rho$, for every $i$ inside the while–loop, thus Lemma C.4 yields

$$\mathbb{E}[B_i|F_{i-1}] \leq \mathbb{E}[G_i|F_{i-1}].$$

This implies that

$$\mathbb{E}[|F| \mid \mathcal{E}] \leq 2 \sum_i \mathbb{E}_{F_{i-1}}[\mathbb{E}[G_i|F_{i-1}]]|\mathcal{E}] \leq 2k^*, \tag{7}$$

where the last inequality follows from the definition of $G_i$, as $\sum_i G_i \leq |F^*| = k^*$.

We continue with the third clause of the theorem. Let $M$ denote the sample complexity of the algorithm, namely $M = M_F + M_E$ where $M_F$ is the expected total number of trajectories sampled within the FINDING SAFEZONE Algorithm (without the samples made by *EstimateSafety* Subroutine) and $M_E$ is the total number of trajectories sampled using *EstimateSafety*. We will bound each term separately.

Since $\mathcal{E}$ holds, whenever we are inside the while–loop, $\Delta(F_i) \geq 2\rho$, which implies that it takes at most $1/2\rho$ trajectories in expectation to sample a trajectory that escapes $F_i$, and such trajectory is accepted with probability at least $1/H$.

Thus, from Wald's identity, it follows that

$$\mathbb{E}\left[M_F | \mathcal{E}\right] = \frac{H}{2\rho} \cdot \mathbb{E}[|F| \,|\mathcal{E}] \leq \frac{Hk^*}{\rho}.$$

From Markov's inequality on the above inequality, with probability at least $1 - \frac{\lambda}{4}$,

$$\Pr\left[M_F \geq \frac{4Hk^*}{\rho\lambda}|\mathcal{E}\right] \leq \frac{\lambda}{4}. \tag{8}$$

Moving on to bound $M_E$. Since $\mathcal{E}$ holds, it follows from Equation (7) and Markov's inequality that

$$\Pr\left[|F| \geq \frac{8k^*}{\lambda} \mid \mathcal{E}\right] = \Pr\left[|F| \geq 2k^* \cdot \frac{4}{\lambda} \mid \mathcal{E}\right] = \Pr\left[|F| \geq \mathbb{E}[|F| \mid \mathcal{E}] \cdot \frac{4}{\lambda} \mid \mathcal{E}\right] \leq \frac{\lambda}{4}. \tag{9}$$

If $|F| \leq \frac{8k^*}{\lambda}$, the number of calls for Subroutine *EstimateSafety* is also bounded by $8\pi k^*/\lambda$ (we only call *EstimateSafety* after we added states to $F$). It also implies that $\frac{3\lambda^3}{2(8\pi k^*)^2} \leq \lambda_j$ for every $j \geq 1$. Thus, if $|F| \leq \frac{8k^*}{\lambda}$,

$$M_E = \sum_{j=1}^{|F|} N_i \leq \sum_j^{\frac{8k^*}{\lambda}} \frac{1}{2\epsilon^2} \ln \frac{2}{\lambda_j} \leq \sum_j^{\frac{8k^*}{\lambda}} \frac{1}{2\epsilon^2} \ln \frac{2}{\frac{3\lambda^3}{2(8\pi k^*)^2}} \leq \sum_j^{\frac{8k^*}{\lambda}} \frac{1}{2\epsilon^2} \ln \frac{86(\pi k^*)^2}{\lambda^3}$$

$$= \frac{8k^*}{2\lambda\epsilon^2} \ln \frac{86(\pi k^*)^2}{\lambda^3} = \frac{4k^*}{\lambda\epsilon^2} \ln \frac{86(\pi k^*)^2}{\lambda^3}$$

Combining the above with Equation (9), we get

$$\Pr\left[M_E > \frac{4k^*}{\lambda\epsilon^2} \ln \frac{86(\pi k^*)^2}{\lambda^3} \mid \mathcal{E}\right] \leq \frac{\lambda}{4} \tag{10}$$

As $M = M_F + M_E$, union bound over Equation (6), Equation (8) and Equation (10) implies that with probability $\geq 1 - 3\lambda/4 > 1 - \lambda$,

$$M = O\left(\frac{k^*}{\lambda\epsilon^2} \ln \frac{k^*}{\lambda} + \frac{Hk^*}{\rho\lambda}\right) \tag{11}$$

For each trajectory we sample we run in time $O(H)$, e.g., by using a lookup table for maintaining the current set $F$. Consequently, if the event in Equation (11) holds then the running time of the algorithm is bounded by

$$O\left(\frac{Hk^*}{\lambda\epsilon^2}\ln\frac{k^*}{\lambda} + \frac{H^2k^*}{\rho\lambda}\right).$$

Overall, all the clauses in the lemma hold with probability $\geq 1 - \lambda$.

$\square$

## C.2 Proof of Theorem 4.2

**Theorem 4.2.** *(main theorem) Given $\epsilon, \rho, \delta > 0$, if we run* FINDING SAFEZONE *for $\Theta(\frac{1}{\delta})$ times and return the smallest output set, $F \subseteq \mathcal{S}$, then with probability $\geq 0.99$*

1. *The escape probability is bounded by $\Delta(F) \leq 2\rho + 2\epsilon$.*

2. *The size of $F$ is bounded from above by $|F| \leq (2+\delta)k^*$.*

3. *The total sample complexity and running time are bounded by $O(\frac{k^*}{\delta^2\epsilon^2}\ln\frac{k^*}{\delta} + \frac{Hk^*}{\rho\delta^2})$, and $O(\frac{Hk^*}{\delta^2\epsilon^2}\ln\frac{k^*}{\delta} + \frac{H^2k^*}{\rho\delta^2})$, respectively.*

*Proof.* Assume we run FINDING SAFEZONE Algorithm for $m = \frac{2\ln 300}{\delta}$ times and denote each algorithm output by $F^i$. Return the smallest set $F = \operatorname{argmin}_{F^i}|F^i|$.

It follows from Theorem 4.1 that for every $\lambda \in (0, 1)$, each $F^i$ is of expected size $\mathbb{E}[|F^i|] \leq 2k^*$, and is $(2\rho + 2\epsilon)-$safe with probability $\geq 1 - \lambda$. Choosing $\lambda = \frac{0.01}{3m}$ implies

$$\Pr[\Delta(F) > 2\rho + 2\epsilon] \leq \frac{0.01}{3}. \tag{12}$$

In addition, from Markov's inequality it follows that for every $\delta > 0$,

$$\Pr\left[|F^i| > (2+\delta)k^*\right] \leq \Pr\left[|F^i| > (2+\delta)k^*|\mathcal{E}\right] + \Pr[\mathcal{E}]$$

$$\leq \frac{2k^*}{(2+\delta)k^*} + \lambda$$

$$= 1 - \frac{\delta/2}{1+\delta/2} + \lambda$$

$$= 1 - \frac{\delta/2 - \lambda - \lambda\delta/2}{1+\delta/2}$$

From the independence of the algorithm runs, for $m = \frac{2\ln 300}{\delta}$,

$$\Pr[|F| > (2+\delta)k^*] \leq \Pr[\forall i : (|F^i| > (2+\delta)k^*)]$$

$$\leq \prod_{i\in[m]} \Pr[|F^i| > (2+\delta)k^*]$$

$$\leq \left(1 - \frac{\delta/2 - \lambda - \lambda\delta/2}{1+\delta/2}\right)^m$$

$$\leq e^{-m(\frac{\delta/2-\lambda-\lambda\delta/2}{1+\delta/2})} \leq \frac{0.01}{3}.$$

Hence

$$\Pr[|F| > (2+\delta)k^*] \leq \frac{0.01}{3}. \tag{13}$$

As for the sample complexity, let $M_i$ denote the (random) sample complexity of the $i-$th run, and let us denote

$$\bar{M} = \frac{4k^*}{\lambda\epsilon^2}\ln\frac{86(\pi k^*)^2}{\lambda^3} + \frac{4Hk^*}{\rho\lambda}.$$

From Theorem 4.1, $M_i > \bar{M}$ with probability $< \lambda$.

By taking the union bound on the sample complexity bound per one run, we get

$$\Pr\left[\exists i : M_i > \bar{M}\right] \leq \sum_{i \in [m]} \Pr\left[M_i > \bar{M}\right] \leq m \cdot \lambda = \frac{0.01}{3}.$$

Where the last inequality follows from Theorem 4.1, and $\lambda = \frac{0.01}{3m}$.

Assigning $m = \frac{2\ln 300}{\delta}$ and $\lambda = \frac{0.01}{3m} = \frac{0.01\delta}{6\ln 300}$, we get that with probability $\geq 1 - \frac{0.01}{3}$,

$$\sum_{i=1}^{m} M_i = O\left(\frac{mk^*}{\lambda\epsilon^2}\ln\frac{k^*}{\lambda} + \frac{mHk^*}{\rho\lambda}\right) = O\left(\frac{k^*}{\delta^2\epsilon^2}\ln\frac{k^*}{\delta} + \frac{Hk^*}{\rho\delta^2}\right) \tag{14}$$

Since the algorithm runs in time $O(H)$ for every trajectory sampled, if the sample complexity is bounded by the above term, then the total running time is bounded by $O\left(\frac{Hk^*}{\delta^2\epsilon^2}\ln\frac{Hk^*}{\delta} + \frac{Hk^*}{\rho\delta^2}\right)$.

Finally, from union bound over Equation (12), Equation (13) and Equation (14) all the theorem properties hold with probability $\geq 0.99$. $\qquad\square$

## D  Additional Figures for Section 5

### D.1  Comparing SAFEZONE of two policies

In this section, we empirically explore the SAFEZONE of two different policies within the same MDP. The first policy, described in the previous section, first goes right and then to the middle, and the second policy first goes to the middle and then goes right. See Figure 6 in the appendix. These seemingly similar policies induce very different SAFEZONES as can be seen in Figure 8 which depicts the number of visits in each state. It shows that the second policy requires fewer states to achieve the same level of safety, even though in terms of minimizing the number of steps to get to the goal state it is outperformed by the first policy (intuitively, the second policy has more fail attempts to go up in expectation since the lowest row of the grid cannot get worst). In Figure 7 we see that already with 14% of the states, all three algorithms achieve trajectory coverage of more than 85%.

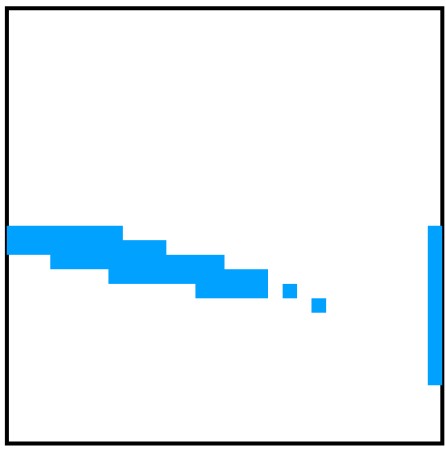

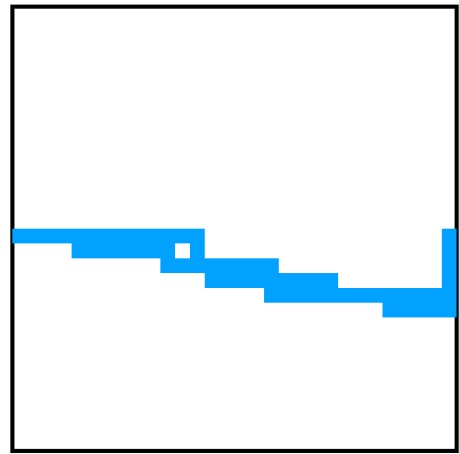

(a) Set chosen by GREEDY AT EACH STEP Algorithm.

(b) Set chosen by SAFEZONE Algorithm.

Figure 5: Empirical results regarding Coverage of the different algorithms, FINDING SAFEZONES and state visit frequency.

Figure 6 depicts the two policies discussed in the paper when $N = 7$.

Figure 7 depicts coverage percentage for the different algorithms discussed in the paper when applied to the second policy.

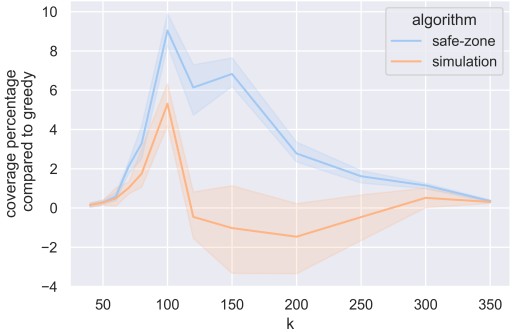

(a) %Coverage: difference from GREEDY Algorithm.

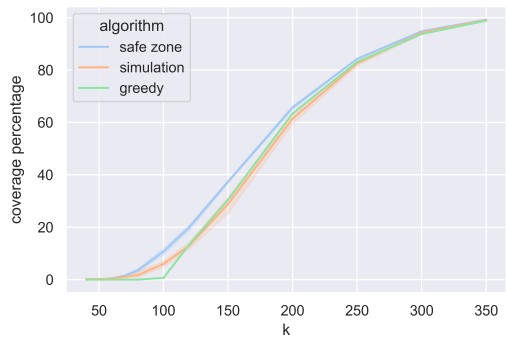

(b) %Coverage: absolute values.

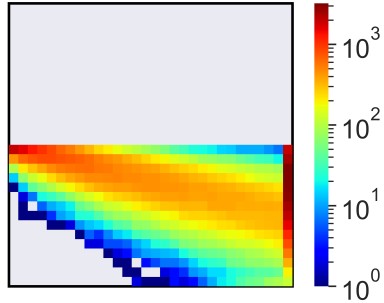

(c) Total number of visits at each state
from 2000 episodes. Zero visits in grey.

Figure 6(c) depicts the number of total visits at each state using the described policy.

Figure 8 shows the visits of the policies described in the main paper for $N = 30$. It is immediately clear that the SAFEZONE of the two policies are fundamentally different. As mentioned, this affects their SAFEZONE sizes. Namely, when trying to go right from a current state in the lowest row it is impossible to get to a square that is lower than that, and the first policy takes advantage of this. In contrast, the second policy keeps trying to go up from the lowest row, which implies that in expectation it goes down more times compared to the first.

# E Exact Computation

In this section, we assume that the transition function is known to the algorithm and show how to compute $\Delta(F)$.

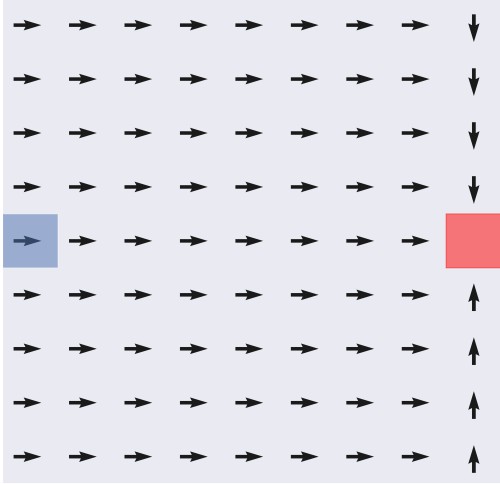

(d) Go right and then to the goal state.

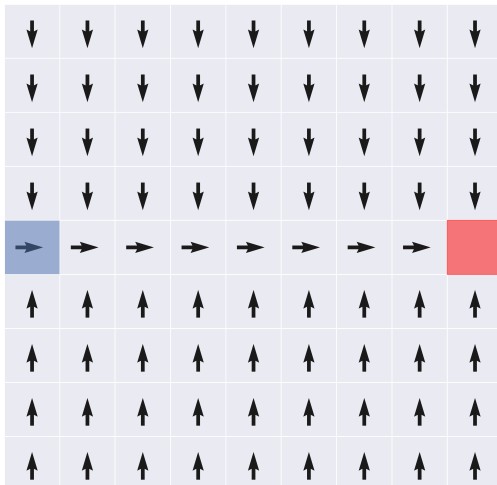

(e) Go to the middle and then right.

Figure 6: Two policies for the same MDP with $N = 7$. Starting state, $s_0$, in blue, and the goal state in red.

Given a Markov Chain $\langle \mathcal{S}, P, s_0 \rangle$ and a set $F \subseteq \mathcal{S}$ we create a new Markov Chain $\langle \mathcal{S}', P', s_0 \rangle$ as follows. We add a new state $s_{sink} \notin \mathcal{S}$, and set $\mathcal{S}' = F \cup \{s_{sink}\}$. For each transition from a state $s \in F$ to a state $s' \notin F$ we modify and make the transition in $P'$ to the sink $s_{sink}$. In $P'$, when we are in $s_{sink}$ we always stay in $s_{sink}$. More formally: (1) if $s, s' \in F$ then $P'(s'|s) = P(s'|s)$, (2) we set $P'(s_{sink}|s) = \sum_{s' \notin F} P(s'|s)$ and (3) $P'(s_{sink}|s_{sink}) = 1$ and $P'(s|s_{sink}) = 0$ for $s \neq s_{sink}$.

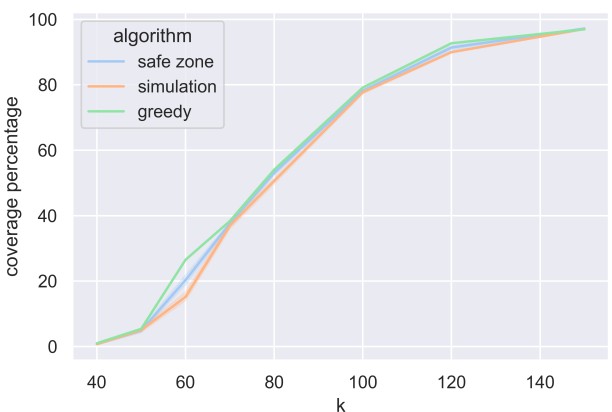

Figure 7: SAFEZONE coverage for the second policy.

Now we claim that $\Delta(F) = \Pr_{P'}[s_H = s_{sink}]$, since any trajectory that reaches a state not in $F$ will reach the sink in $P'$ and stay there. We can compute $\Pr_{P'}[s_H = s_{sink}]$ using standard dynamics programming.

The running time of constructing $\langle \mathcal{S}', P', s_0 \rangle$ is $O(|\mathcal{S}|^2)$. Computing the probability of $\Pr_{P'}[s_H = s_{sink}]$ takes $O(H|\mathcal{S}|^2)$. Therefore we have established the following.

**Lemma E.1.** *Given a Markov chain $\langle \mathcal{S}, P, s_0 \rangle$ and a set $F \subseteq \mathcal{S}$ we can compute $\Delta(F)$ in time $O(|\mathcal{S}|^2 H)$.*

Note that the above lemma implements an exact version of the $EstimateSafety$ Subroutine.

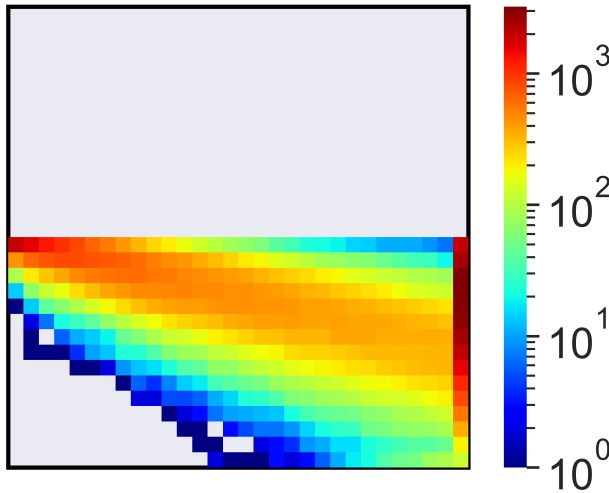

(a) Number of visits at each state for policy "Go right and then to the middle"

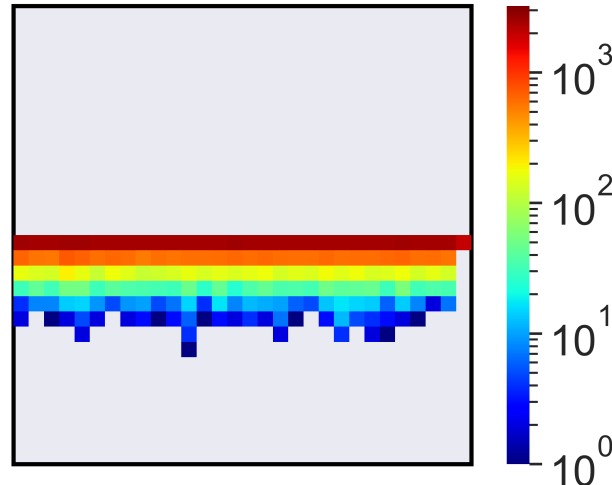

(b) Number of visits at each state for policy "Go to the middle and then right"

Figure 8: Total number of visits for the two policies.

