# OpenReview forum: "Finding Safe Zones of Markov Decision Processes Policies"
_NeurIPS.cc/2023/Conference — NeurIPS 2023 poster_

### Official Review · Reviewer_N6Et · 2023-07-06

**Soundness:** 3 good
**Presentation:** 2 fair
**Contribution:** 3 good
**Rating:** 5
**Confidence:** 3

**Summary:**


The work introduces the SafeZone problem for safe reinforcement learning (RL). Instead of learning for the optimal policy under the constrained Markov Decision Process (MDP) as many traditional methods deal with a safe RL problem, the work attempts to search for a SafeZone, a subset of state space that is policy-dependent, balancing between minimizing the number of states contained in the subset and reducing the probability that a random trajectory goes out of the subset. If the probability of escaping the SafeZone is low with a randomly sampled trajectory, it is considered safe. The paper also proposes a new algorithm for detecting the SafeZone. The algorithm uses rejection sampling to decide whether a new random trajectory should be added to the SafeZone and keep updating the safety estimation once a new trajectory is added.



**Strengths:**

1. The work proposes a new way of dealing with the safe reinforcement learning (RL) problem. In traditional safe RL, the problem is usually formulated as a constrained MDP, while in this work, the agent is trained to find the SafeZone for a given policy.

2. The paper detailedly explains the formalization of the new problem setting and compares the new setting with the constrained MDP, it helps with understanding.

3. The paper provides both theoretical analysis and empirical evidence, making the results more sound. Though solving the SafeZone problem is NP-Hard, the paper derives an algorithm and provides empirical results. These results suggest that the idea introduced in the paper is practical.


**Weaknesses:**

The main concern comes from the choice of baseline algorithms and the experiment design. The paper empirically tested methods on a $N \times N$ grid, checking the quality of the SafeZone given by the new algorithm and baselines. A SafeZone with a smaller size but a larger coverage percentage on the trajectory (lower escape probability) is considered better. Although the new method is empirically shown to be better than the baseline algorithms regarding the above standard, I doubt whether the proposed algorithm can be sufficiently tested by the reported experiment, or, whether the advantage and shortage of SafeZone problem setting can be seen from this experiment.

In the introduction section, the paper points out that a new method is proposed for safe RL. One common idea used by the community is to formulate the problem as constrained MDP to solve the safe RL problem, as the paper points out. The proposed SafeZone method performs as the other path for solving safe RL, besides using constrained MDP. So, one thing remaining unclear is whether SafeZone is a better way of solving the safe RL problem than constrained MDP.

The paper focuses on comparing the quality of SafeZone detected by the proposed algorithm and baselines, but does not provide any empirical evidence about whether a good SafeZone output helps with improving the learning performance in safe RL problems, or whether a SafeZone method performs better than constrained MDP tasks. As SafeZone is a newly introduced problem setting and the proposed algorithm directly targets searching for a better SafeZone, I am not surprised that the new method outperformed baselines regarding the SafeZone quality. But the lack of learning performance in safe RL tasks makes it hard to say if detecting a SafeZone is a better way of solving the safe RL problem. It could be more convincing if the learning performance in safe RL tasks could be checked and empirically indicate that the SafeZone idea can learn safe RL tasks more efficiently than using the constrained MDP.



**Questions:**

My question is related to the concerns listed above. I would appreciate it if the authors could discuss whether the proposed method has the potential to outperform the constrained MDP methods, regarding the learning efficiency.


**Limitations:**

Yes

---

> ### Author Rebuttal · Authors · 2023-08-09
>
> We thank the reviewer for the detailed review!
>
> Our focus is theoretical, the experiments are a minor part of the paper and were only made for demonstration.
>
> We don't offer an alternative solution to constrained MDP; instead, we approach a different problem and give a different solution.  As we explained in the introduction, the purpose of the SafeZone problem is to capture a new sort of safety and aims to capture popular events in the environment.

---

> > ### Comment · Reviewer_N6Et · 2023-08-18
> > **Reply to the rebuttal**
> >
> > Thank you for your reply. After reading it, I intend to maintain my original score.

---

### Official Review · Reviewer_EfcG · 2023-07-06

**Soundness:** 2 fair
**Presentation:** 1 poor
**Contribution:** 2 fair
**Rating:** 5
**Confidence:** 3

**Summary:**

This paper introduces the SafeZone problem: Given an MDP and a policy, find the smallest set of states such that the probability of leaving this set of states (the escape probability) lies below a given threshold using trajectory samples. The authors provide various examples of applications of the SafeZone problem, such as imitation learning with compact policy representation and post hoc explainability of RL.
The authors provide proof that the SafeZone problem is NP-complete. The adjusted problem solved in this paper is to instead find a SafeZone with a larger but minimal escape probability than the threshold and a larger but minimal cardinality than the true optimal cardinality given the threshold.
Three naive approaches are discussed, and one approximation algorithm for finding SafeZones. The authors provide upper bounds on each approach's escape probability and sample complexity. They give provable guarantees on their approximation algorithm. Finally, the paper compares three of the four approaches based on a grid world problem.

Part of this work has been previously presented at the NeurIPS 2022 TSRML Workshop. The work did not appear in any proceedings, journals, or books, so according to NeurIPS' call for papers, this is not considered a dual submission.

**Strengths:**

The paper introduces a novel problem and gives extensive examples of situations where a solution to this problem could be useful. It shows the need for approximation by proving that the SafeZone problem is NP-complete, even if the induced MC and the minimal cardinality (of the corresponding escape probability threshold) are known.

The paper analyses four algorithms for finding SafeZones, provides upper bounds on the escape probability and sample complexity, and indicates the limitations of the three naive approaches.

The paper suggests an interesting future work direction, where the problem is moved from finding SafeZones of an induced MC to finding policies of an MDP with small SafeZones given an escape probability threshold.

**Weaknesses:**

The paper is challenging to follow because of two main reasons: a lot of information is only available in the appendices, and few (intuitive) examples are used after introducing the problem.

The first reason, limited information in the paper itself and reliance on the appendices can, for example, be seen in Section 3, where Appendix B contains the actual algorithms, MDP examples, and proof outlines, and in Section 5, which only contains one figure, with the rest all placed in Appendix E.
Furthermore, what information is available in the appendices is also not always clear. For example, it is not mentioned in Appendix A that the proof of Theorem A.2 is given in Appendix D. Nor is it mentioned in Section 5 that more figures regarding the empirical demonstration are available in Appendix E, only four subfigures are referenced.
Moreover, the information about the empirical demonstration section is incomplete. The escape probability threshold is not given for computing the results and the number of repetitions. Also, the paper only compares three of the discussed methods without mentioning the exclusion of the Greedy by Threshold algorithm.

The second reason, the lack of intuitive examples, concerns the sections after the introduction. The paper gives extensive and intuitive examples in the introduction but does not use these throughout the rest of the paper. Only the autonomous vehicle example is referenced once in Section 2.
Although the paper proves why an approximation of the SafeZone problem is necessary, they give no argumentation as to why an almost 2 approximation is sufficiently tight to have a practical use still. An intuitive example where the size of the SafeZone is compared to an optimal SafeZone could illustrate this usefulness.
Also, the problem used in the empirical demonstration does not provide any intuition regarding the applications of the SafeZone problem, as given in the introduction. Some discussion on how this relates would be helpful.

Typo in Section 3, line 208. theses -> these
Incorrect reference in Section 5, line 355. Section 5 -> Figure 1.

**Questions:**

Why is Greedy by Threshold excluded from the empirical demonstration?

How should an escape probability threshold be chosen?

What escape probability threshold value was used for the empirical demonstrations?

How often were the experiments of the empirical demonstration repeated?

---

> ### Author Rebuttal · Authors · 2023-08-09
>
> Thank you for the helpful and detailed review. In what follows we address your comments:
>
> **Proof of Theorem A.2 is given in Appendix D**
> We apologize for the confusion regarding the proof of Theorem A.2 and we will mention that it appears in Appendix D and that there are more figures in App E in the final version of the paper.
>
> **How should an escape probability threshold be chosen?**
> A feature of the algorithm is that the escape probability is a free parameter that can be tuned depending on the use case, or experimented to select the ideal one for the target (e.g., using binary search).
>
> **Escape probability threshold in demonstration**
>  Rather than stopping the algorithm when it reaches some escape probability threshold, the implementation adds trajectories one after another until reaching a specific safezone size (the reason being that the x-axis is k).
>
> **Greedy by threshold**
> We only implemented one type of greedy algorithm, as the MDP in the demonstration is not layered (i.e., there are states that appear in more than a single time step, see Line 220).
>
> **Intuitive examples**
> We appreciate the advice and will use the autonomous vehicle example as a running example throughout the paper.
>
> **How often were the experiments of the empirical demonstration repeated?**
>  We ran each experiment 2000 times (see line 353).

---

> > ### Comment · Reviewer_EfcG · 2023-08-19
> > **Thank you.**
> >
> > I thank the authors for their responses, these clarify my questions. I will keep my score.

---

### Official Review · Reviewer_JjfH · 2023-07-07

**Soundness:** 3 good
**Presentation:** 2 fair
**Contribution:** 3 good
**Rating:** 5
**Confidence:** 3

**Summary:**

The paper introduces a new problem that involves finding a SafeZone: given the possibility to interact with an MDP using a fixed policy, the learner as to find a subset of states F of minimal size s.t. the probability that a trajectory visits a state outside F is at most a given parameter \rho. They show that the problem is NP-hard even when the transition matrix is known. They propose an approximate algorithm that finds near-optimal SafeZones with polynomial sample complexity (in the relevant variables). The approach is numerically tested on a toy problem.

**Strengths:**

1. The paper introduces an interesting problem (though it may lack a bit of "concrete motivation", see below). The problem is definitely challenging from a theoretical perspective.
2. While I haven't checked the proofs, the results seem sound
3. The authors managed to derive a computationally-efficient algorithm computing near-optimal Safe Zones even though the base problem is NP-hard

**Weaknesses:**

1. Although the introduction presents some examples, I still wasn't fully convinced about the motivations behind studying this peculiar setting. In particular, it is still unclear to me what problems the proposed algorithm (or in general finding safe zones) enable solving. It would be good to provide a more "concrete" example, ideally some numerical results on a real-world problem (or a simulated variant) where it is clear that finding safe zones is the right thing to do
2. Moreover, even in the context of the examples given in the introduction, the assumption that we can only interact with the MDP with a single policy seems a bit limiting. In those settings, it seems reasonable to observe trajectories collected by multiple policies, mostly because multiple agents are interacting with the environment and each displays learning behavior (so changing policies). How to extend the proposed setting and algorithms to such a context is not clear
3. I found the core part of the paper (Sec. 3,4) a bit hard to read. In particular, if my understanding is correct, Sec. 3 tries to build some intuition which is later useful to explain the main algorithm in Sec. 4. However, that didn't really work for me: Sec. 4 starts with "In this section, we suggest a new algorithm that builds upon and improves the added trajectory selection of the SIMULATION Algorithm", but the SIMULATION algorithm was not explained at all in Sec. 3. The rest of Sec. 4, especially the first paragraphs, are also hard to follow, while Sec. 3 seems mostly to report a bunch of technical results rather than intuitions (so I did not really feel "a gentle start" to me).
4. It is not clear how good the sample complexity of the main algorithm (Th. 4.2) is, especially in terms of dependences on the main variables (k^\star, \delta). It would be good to add some discussion about this. Do the authors think that the current dependences are optimal or improvable?
5. The experiments are conducted on a very toy and low-scale domain. I would have like to see larger domains and also a comparison of the computational requirements (time complexity) of the different algorithms.

**Questions:**

See above.

**Limitations:**

I did not found any discussion about limitations

---

> ### Author Rebuttal · Authors · 2023-08-09
>
> Thank you for the helpful and detailed review. We will address your comments below.
>
> **MDP with a single policy**
>
> Our setting does allow for any number of policies (multiple agents) by using a  single mixed policy as follows: each agent $i\in\{1,\ldots,n\}$ has some policy $\pi^i$. The mixed policy is the one that selects a number uniformly at random from  $1,\ldots,n$, then runs $\pi^i$ (alternatively, we can consider any other distribution over the agents/policies). As a result, it is possible to observe trajectories collected by multiple policies and use the algorithms from the paper as is. Thank you for bringing it up! We will discuss it in the final version of the paper.
>
> **Sections 3+4 readability**
>
> SIMULATION is an intuitive algorithm that simply samples a certain amount of random trajectories and adds their states to a set, and then returns this set. It is described in line 207 and formalized in Appendix B as Algorithm 4. If anything is unclear regarding SIMULATION Algorithm 4 or Sections 3,4 we would be happy to answer further questions.
>
> **Sample complexity of the main algorithm (Th. 4.2)**
>
> We expect the dependency on $k^\star$ to be optimal. The reason why is the following. Consider a MC such that there are $k^*-1$ trajectories, each starts in an initial state $s^0$ and ends with a unique corresponding state $1,\ldots,k^*-1$. Consider a case where the probability of each such trajectory is $\frac{1-\delta}{k^*-1}$. In this case, it would take at least $k^*$ samples to find a $k^*$-safezone.
> As for the parameter $\delta$, this could be treated as a constant. For example, selecting $\delta=1/3$ yields that we need to run the algorithm $6\cdot \ln 300$ times and a solution of size $7/3k^*$ w.h.p..
>
> We will add a discussion about this in the final version of the paper.
>
> **Experiments + Time complexity of the naive algorithms**
>
> As we were the first to formalize the safezone problem, we focused on theoretical guarantees and provided some experiments just for demonstration.
> As for time complexity of the naive algorithms:
> - In Greedy by Threshold, the running time is bounded by the maximum between the number of states reachable by the policy with probability $>0$ , i.e., by $|S|$.
> - Simulation Algorithm has a running time of $O(H/\beta \ln k^∗)$ as it samples $O(1/\beta \ln k^∗)$ trajectories and each has at most $H$ states.
> - In Greedy at Each Step, as there are MC with states reachable in every level, and it needs to rank each level, it has a running time of $O(H|S|\log|S|)$ (using, e.g., mergesort as sorting algorithm).
>
> **Motivation**
>
> The motivation of our paper, as discussed in the introduction and discussion sections, emphasizes the practical significance of our approach. By identifying SafeZones, we offer solutions to challenges in autonomous vehicles, manufacturing, and potentially compact policy design.

---

> > ### Comment · Reviewer_JjfH · 2023-08-15
> > **Response to Rebuttal**
> >
> > Thank you for the detailed response. The fact that the proposed algorithms can straightforwardly deal with multiple policies is quite interesting and should be definitely mentioned in the paper. I have increased my score accordingly. I still think that the paper should be improved in terms of clarity and motivation. Maybe, as Reviewer EfcG suggests, using one of the practical use cases given in the introduction as a running example throughout the whole paper could help in both these aspects.

---

> > > ### Author Response · Authors · 2023-08-16
> > >
> > > Thank you for your quick response and for updating your score!
> > > We appreciate your helpful suggestions. We will incorporate the explanation that multiple policies can be dealt with, along with the practical example of autonomous driving, throughout the paper.

---

### Official Review · Reviewer_YYwR · 2023-07-17

**Soundness:** 3 good
**Presentation:** 2 fair
**Contribution:** 3 good
**Rating:** 6
**Confidence:** 3

**Summary:**

The paper proposes an innovative definition called the SAFEZONE and uses it to describe the escape probability of the sampled trajectory. This paper analyzes several naïve algorithms and proposes an algorithm to overcome the weaknesses in naïve algorithms, especially when considering the size of safety states. Finally, numerical experiments are conducted to evaluate the algorithm's performance compared to mentioned naïve algorithms.

**Strengths:**

I think the paper is easy to follow.

Strengths:

1. The paper introduces the definition called the SAFEZONE to find the subset of ‘safe states’, i.e., the set that has low escape probability $\rho$ and a small size $k$.
2. Three naïve algorithms are clearly discussed with solid theoretical analysis and specific MDPs examples in the appendix.
3. A new approach for solving an approximate SAFEZONE is proposed with complete analysis, and numerical comparisons show the algorithm's performance.

**Weaknesses:**

No obvious limitations, but this paper is not written well. The motivation for this work should be discussed more, and it would be helpful if the author discuss more concurrent works.

**Questions:**

No more questions.

**Limitations:**

No.

---

> ### Author Rebuttal · Authors · 2023-08-09
>
> We appreciate your evaluation of our paper. We will address your valuable suggestions and enhance the clarity and motivation of our paper. We are grateful for your positive comments on our paper's concepts and algorithms.
>
> The motivation of our paper, as discussed in the introduction and discussion sections, emphasizes the practical significance of our approach. By identifying SafeZones, we offer solutions to challenges in autonomous vehicles, manufacturing, and potentially compact policy design.
>
> We will discuss more concurrent works (e.g., about other approaches for RL safety) to provide a comprehensive context for our contributions as you suggested.

---

### Author Rebuttal · Authors · 2023-08-09

We would like to thank the reviewers for their valuable and informative reviews! We are glad you appreciate the new ֿSafeZone definition and our theoretical contribution. If we misunderstood any of the questions, we would be happy to clarify any further information during the discussion period.

---

### Comment · Area_Chair_rZHz · 2023-08-18

I would like to thank the authors for providing detailed responses to all referee reports, and I apologize that some of the referees have not responded to the rebuttals despite multiple reminders. I will account for that in my final recommendation.

---

> ### Author Response · Authors · 2023-08-21
>
> We appreciate your consideration and look forward to your final recommendation.

---

### Decision · Program_Chairs · 2023-09-21

**Decision:**

Accept (poster)

**Comment:**

Overall, I am suggesting to accept this paper. The reviewers consistently value the contributions of the paper, as well as the soundness of the analysis, and I agree with them. That said, the review team has also brought up a number of consistent concerns regarding the motivation (including the lack of clearly motivated and easy to follow examples) as well as the writing of the paper. I sincerely hope that the authors will follow through with their promises and improve upon those parts.